# Signatures of the sub-Rayleigh to supershear fracture transition in snow avalanche experiments

Bastian Bergfeld [1], Johan Gaume [1,2,3] ✉, Gregoire Bobillier[1], Alexandre Pellet [3,4], Jürg Schweizer [1] & Alec van Herwijnen [1]

Snow slab avalanches occur when a crack propagates within a highly porous weak snow layer buried beneath a cohesive snow slab. Here, we report direct observations of a supershear event in snow fracture experiments following the spontaneous transition from sub-Rayleigh to intersonic crack propagation. The experiments involve artificially triggered avalanches on a small slope with a natural snowpack, captured with high-speed cameras and analyzed using digital image correlation. Deformation fields reveal distinct signatures: slope-normal collapse of the weak layer and slab flexure drive sub-Rayleigh propagation, while supershear fracture is related to slope-parallel deformation and slab tension. These results are further reinforced by numerical simulations that replicate the experiment and provide strong supporting evidence that the Burridge-Andrews mechanism governs the transition to supershear propagation. Analogous to supershear strike-slip earthquakes linked with substantial magnitudes, our findings suggest that supershear avalanches relate to widespread crack propagation and large avalanche dimensions, holding significant implications for risk mitigation strategies.

The release of a dry-snow slab avalanche (Fig. 1a) follows a sequence of failure and fracture processes[1]. An initial failure is initiated in a weak layer of snow buried beneath a cohesive snow slab. This leads to the formation of a crack which, under certain conditions, can propagate across the slope, leading to loss of support and sliding of the slab in steep terrain[2]. The propagation mechanism was initially understood in terms of strain-softening-induced shear fracture[3,4]. As this model did not explain crack propagation in low-angle terrain and remote avalanche triggering, Heierli et al.[5] extended the anticrack concept–initially proposed to explain the failure process in deep earthquakes[6], also suspected in firn and snowquakes[7,8]–to the avalanche release process. Based on experimental evidence[9–12], their model accounted for the volumetric collapse of the weak layer, which leads to closure of crack faces under normal slab loading. This enables the slab to flex, generat-

ing stress concentrations at the crack tip, thus facilitating propagation even in the absence of a driving shear force. In such a propagation mode, the crack speed should theoretically be limited by the Rayleigh wave speed[13]. The debate between the two proposed mechanisms, shear versus anticrack, has sparked intense discussion over the past three decades[14–16].

In 2014, Hamre et al.[17] reported, based on indirect measurements from avalanche videos, fracture speeds exceeding 200 m s$^{-1}$, surpassing the Rayleigh wave speed of the slab, adding a new dimension to the discussion. These findings appeared contradictory, not only exceeding theoretical bounds but also all values obtained from contemporary state-of-the-art fracture tests which consistently yielded sub-Rayleigh speeds below 60 m s$^{-1}$, typically[10,16,18,19]. Recently, Trottet et al.[20] conducted numerical simulations indicating that the latter discrepancy

[1]WSL Institute for Snow and Avalanche Research SLF, Davos, Switzerland. [2]Climate Change, Extremes and Natural Hazards in Alpine Regions Research Center CERC, Davos, Switzerland. [3]Institute for Geotechnical Engineering ETH, Zürich, Switzerland. [4]Univ. Grenoble Alpes, CNRS, INRAE, IRD, Grenoble INP IGE, Grenoble, France. ✉e-mail: johan.gaume@slf.ch

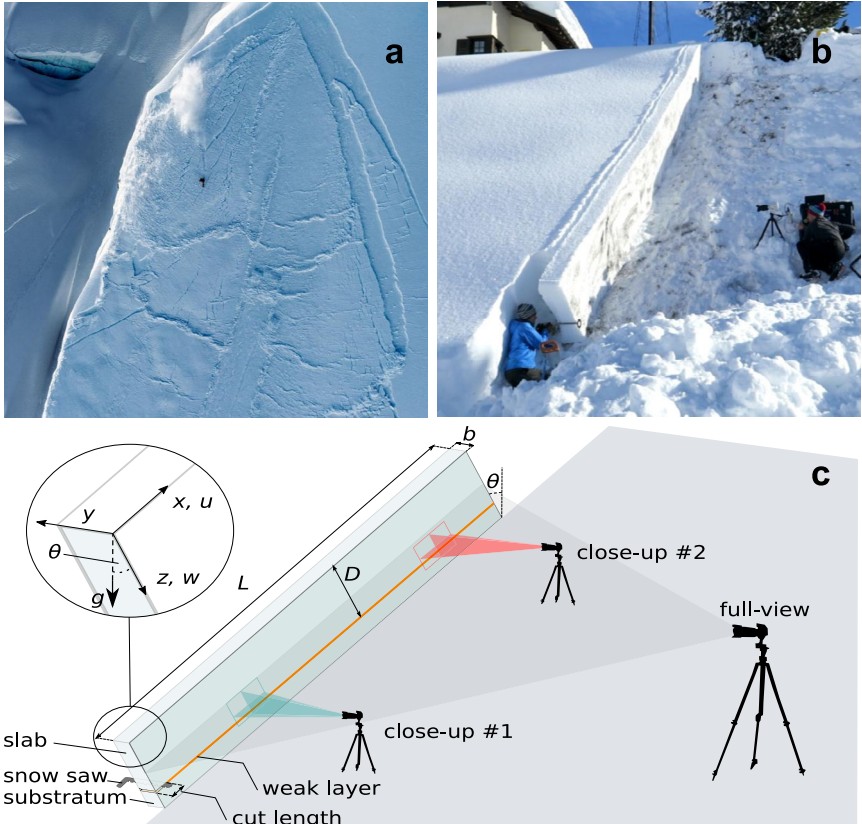

**Fig. 1 | Slab avalanche and setup of the experiements. a** Accidentally triggered snow slab avalanche in Alaska, 2014. Photo: Oli Gagnon. **b** Image of the experimental avalanches consisting of snow fracture tests performed in Davos, Switzerland. **c** Schematic representation of (**b**). The experimental design involves isolating a column with length $L \approx 10$ m and width $b = 0.3$ m. The column height is given by the overall snow height, which is the sum of the thicknesses of substratum, weak layer (orange) and the thickness $D$ of the slab. The column ends were cut slope-normal (zoom top left in **c**). The coordinate system originates at the snow surface edge at the downslope side of the column. $g$ is the gravity vector and $\theta = 37°$ indicates slope angle. Testing is done by using a snow saw to cut uphill into a pre-identified weak layer until the onset of unstable crack growth. Each experiment was recorded with high-speed cameras, either filming a close-up ($\approx 0.6$ m slope-parallel field of view, blue and red) or the full column length $L$ (full-view).

could be explained by a sub-Rayleigh to supershear transition following the Burridge-Andrews mechanism[21,22]. They argue that current state-of-the-art fracture tests were too short to capture this transition towards intersonic fracture speeds. Yet, while this numerical study proposed an indirect validation based on analysis of avalanche videos[23], direct evidence of intersonic–or supershear– fracture in snow avalanches remains non-existent. In particular, the detailed deformation mechanisms occurring around the weak layer are still unknown. The supershear transition in slab avalanches may have crucial implications for the associated risk because it is thought to affect the avalanche release volume and thus its destructive potential[24–26].

To investigate these processes, and inspired by experimental evidence of supershear fracture in laboratory earthquakes[27] and friction experiments[28,29], we designed controlled avalanche experiments on a small snow slope. This real-scale experimental setup allows us to explore the existence and characteristics of supershear fracture in avalanche release under safe and controlled conditions.

In this work, we report direct observations of a transition from sub-Rayleigh to supershear crack propagation. High-speed imaging combined with digital image correlation reveals clear kinematic signatures accompanying this transition, evolving from weak layer collapse and slab flexure during sub-Rayleigh propagation to slope-parallel tension during supershear rupture. Complementary numerical simulations reproduce these dynamics and provide strong evidence that the Burridge-Andrews mechanism governs the onset of supershear propagation.

## Results

### Snow avalanche experiments

Supershear experiments in snow are extremely challenging, as they require a highly unstable snowpack and a safe, accessible slope, conditions that rarely coincide. Even under favorable conditions, long propagation tests rarely succeed, especially for distances of 10 meters or more[30]. The successful experiments presented here followed years of unsuccessful attempts and extensive site preparation to preserve the weak layer and ensure favorable test conditions.

The snow fracture tests involved experimental avalanches triggered on a small snow slope, where a buried weak snow layer was identified in a manual snow profile[31]. The experimental setup adhered to the Propagation Saw Test (PST) standards defined by Greene et al.[32], except using a 10 m sample length (Fig. 1b, c, and Supplementary Movie SM1). This, combined with the steep slope of 37°, ensured that the conditions for supershear fracture as reported in Trottet et al.[20] are met. An artificial crack of increasing size was introduced by sawing into the weak layer until unstable crack growth sets in (Fig. 1b, c). The open sidewall of the PST was speckled with black ink, and a high-speed camera was used to record the speckled wall (full-view perspective). In addition to the full-view, two close-ups focusing on the weak layer were recorded at a distance of about 2 and 8.5 m away from the trigger point (Fig. 1c). Digital Image Correlation (DIC) analysis and camera distortion correction of the high-speed recordings were performed as described by Bergfeld et al.[33]. This allowed us to compute the tangential and normal displacement fields $u(t)$ and $w(t)$, respectively, and to calculate

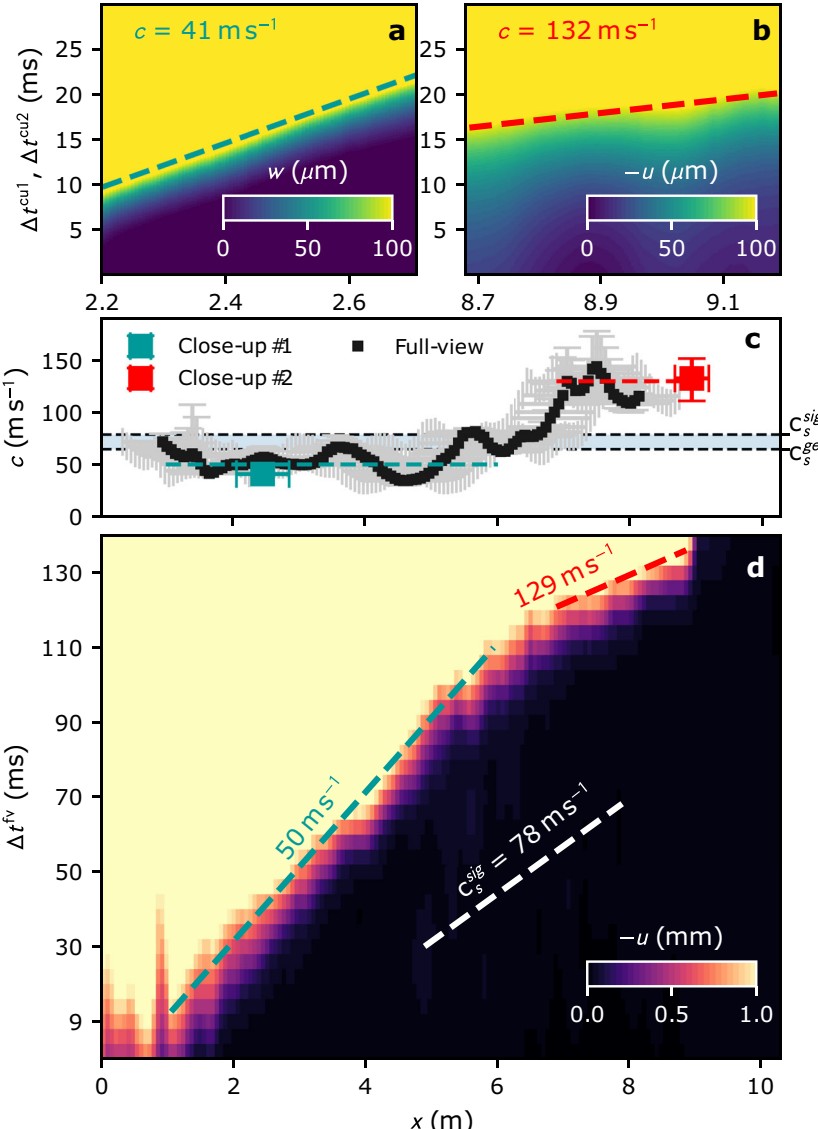

**Fig. 2 | Crack propagation speed in avalanche experiment. a** Normal and (**b**) tangential displacement fields in close-ups #1 and #2, respectively, with associated speed estimates. **c** Comparison of crack propagation speed obtained from the full-view of the PST with the local estimates from the close-ups. **d** The tangential displacement $u$ from the full-view analysis is presented along with the average crack speeds in the sub-Rayleigh and supershear regimes. The slab shear wave speed is included as a reference. Note that very similar crack speed values were obtained at close-up #1 (**a**) based on the tangential displacement $u$. Similarly, from the full-view (**d**), one obtains very similar values of crack speed in the sub-Rayleigh range using the normal displacement $w$ (see Supplementary Data Figs. 5 and 6). $c_s^{sig}$ and $c_s^{ger}$ are the shear wave speed evaluated based on Sigrist's[35] and Gerling's[34] elastic modulus parameterization, respectively (see Methods Section).

crack propagation speeds extending the methodology outlined in Bergfeld et al.[33], as detailed in the Methods Section.

**Supershear fracture**

Three side-by-side fracture experiments were performed at an interval of approximately 2.5 hours. In all tests, the crack propagated the first meters ($x \lesssim 5$ m) at a speed of around 50 m s$^{-1}$ (Fig. 2a, d, Supplementary Figs. 5 and 6), estimated from the three full-view perspectives as well as close-up #1. In PST #1 and PST #2, crack propagation in the weak layer was affected by slab fractures at distances of approximately 5.5 m, from the bottom of the test column (see Methods and Supplementary Fig. 5). In contrast, in PST #3, the absence of slab fractures allowed the crack in the weak layer to transition into a supershear propagation regime (Fig. 2). Based on tangential displacement analysis from the full-scale video, we observed a transition from a crack speed of approximately 50 m s$^{-1}$ to around 130 m s$^{-1}$ after a propagation

distance of about 6 m. We conservatively estimated the slab shear wave speed by using the highest values of existing empirical formulations derived from snow acoustic measurements[34] and dynamic loading[35] (see Supplementary Fig. 7). With these estimates (65 and 78 m s$^{-1}$) and after model-based verification of crack propagation speeds in layered slab systems (see Supplementary Note 3 and Supplementary Fig. 9 therein), we can confidently categorize the measured speed of approximately 50 m s$^{-1}$ as sub-Rayleigh and the measured speed of about 130 m s$^{-1}$ as intersonic, indicating supershear fracture. The analysis of the spatio-temporal evolution of displacement at the close-up location further strengthens this conclusion, offering detailed insights into the deformation mechanisms driving this transition. At the close-up #1, speed evaluations based on both tangential and normal displacements yielded similar crack speeds of around 40 m s$^{-1}$. This supports the hypothesis that, in the sub-Rayleigh regime, the crack propagates as a mixed-mode anticrack[16,20,36]. However, at

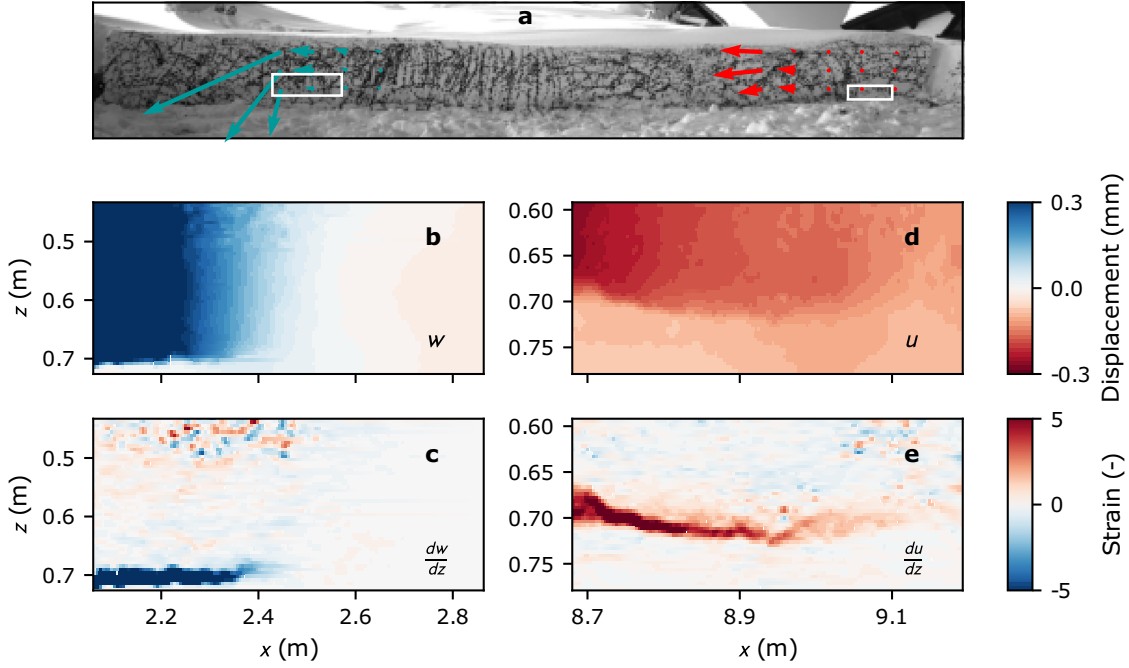

**Fig. 3 | Displacement and deformation analysis.** Spatial patterns of displacement vectors (**a**). Note that the same is provided for PST #1, PST #2 and PST #3 as videos in Supplementary Movies SM5, SM6 and SM7, respectively. Normal displacement (**b**) and strain (**c**) at close-up #1, and tangential displacement (**d**) and shear strain (**e**) at close-up #2 for two time instances corresponding to the passage of the crack tip at the locations of close-up #1 ($\Delta t^{cu1}$ = 14.3 ms) and close-up #2 ($\Delta t^{cu2}$ = 21.8 ms), respectively. The close-ups are identified in (**a**) as white rectangles.

close-up #2, speed evaluation based on tangential and normal displacements differed. Only the analysis based on the tangential displacement led to intersonic speeds as also observed in the full-view (the $w$-based speed was around 80 m/s). This aligns with the assumption[20] that, in this regime, the crack propagates as a shear (mode II) crack, in which volumetric collapse of the weak layer and normal displacement of the slab are uncoupled from the crack tip.

### Deformation mechanisms

Through displacement analysis (Fig. 3), a distinct crack driving mechanism is discerned at each location along the column. At the lower part of the column (close-up #1 location), the predominant mechanism driving crack propagation is slab bending induced by volumetric collapse of the weak layer (see also Supplementary Movie SM2). This is evidenced by the downward-pointing displacement arrows in the full-view analysis (Fig. 3a, blue arrows), the spatial distribution of normal displacement (Fig. 3b), and localized strain patterns, which pinpoint the position of the crack tip (Fig. 3c). It is noteworthy that shear strain analysis did not reveal any discernible localization pattern, in contrast to the normal strain (see Supplementary Fig. 4d). This could be expected since crack propagation in this regime is driven by weak layer collapse and therefore slope-normal displacement.

Conversely, at the upper end of the column (location of close-up #2), both full-view and close-up displacement analyses indicate that the primary crack driving mechanism is slab tension resulting from shear failure of the weak layer (see also Supplementary Movie SM3). This inference is drawn from the slope-parallel orientation of the displacement vectors in the full-view analysis (Fig. 3a, red arrows), as well as the tangential displacement field (Fig. 3d) and the visualization of shear strain localization patterns, which delineate the crack in the weak layer (Fig. 3e). No distinct localization pattern was observed in the normal strain analysis (Supplementary Fig. 5i).

The preceding observations are further clarified by the relative displacement between the forming crack faces by analyzing the

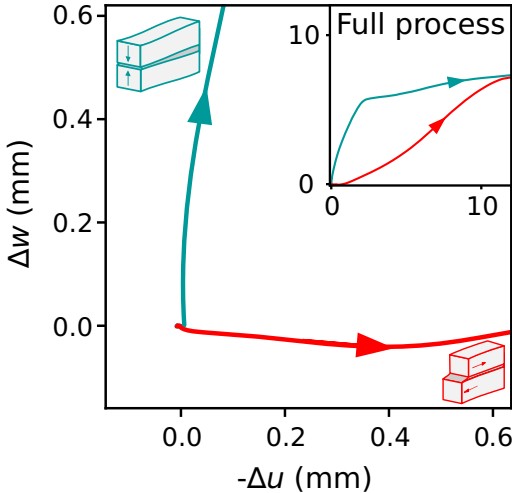

**Fig. 4 | Normal versus tangential displacements.** Relative normal displacement $\Delta w$ with relative tangential displacement $\Delta u$ (relative displacement of the forming crack faces) for close-ups #1 (blue) and #2 (red), highlighting two different deformation mechanisms. Inset: the same plot is displayed with both $x$- and $y$-axis limits extended to larger maximum values, preserving the aspect ratio.

relative normal displacement ($\Delta w$) as function of the relative tangential displacement ($\Delta u$) at the close-up locations (Fig. 4). At close-up #1, the initial displacement predominantly aligns with the slope-normal direction, indicating a crack-closing mechanism (Fig. 4, blue trajectory). Conversely, at the location of close-up #2, the relative displacement primarily aligns with the slope-parallel direction, with a minor negative slope-normal component, indicative of slight dilation (Fig. 4, red trajectory). Note that weak layer collapse still occurred, but later in time, thus representing a secondary process subsequent to supershear fracture (Fig. 4, inset).

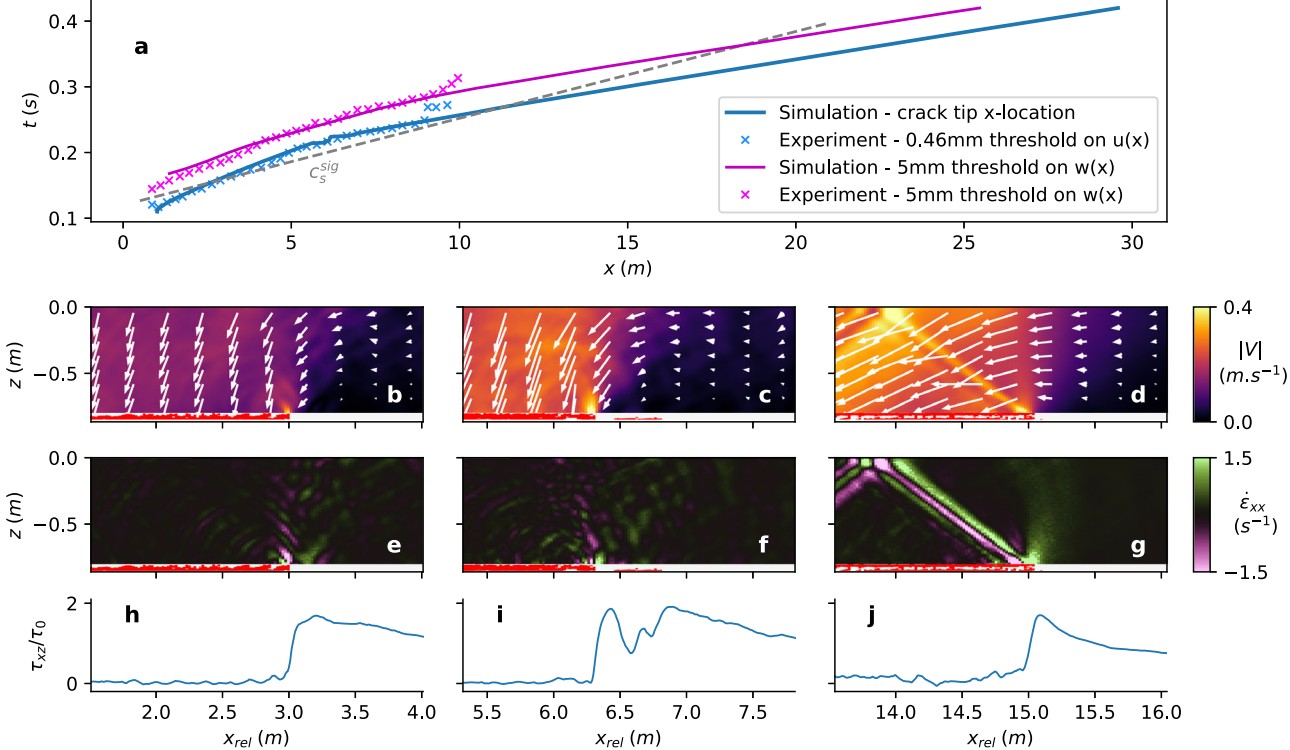

**Fig. 5 | Numerical experiment of weak layer crack propagation beneath a homogeneous snow slab using a 2D Material Point Method. a** Time according to crack tip position, determined by following either the plasticized particles in the weak layer, the slab's slope parallel displacement $u(x)$, or the slab's slope normal displacement $w(x)$. $u(x)$ and $w(x)$ are the average over the slab thickness of $u(x,z)$ and $w(x,z)$ respectively. The grey dashed line represents a virtual crack tip moving at shear wave speed. **b–g** Zoomed views of the slab (color mapped) and the weak layer (white and red) around the crack tip. In the weak layer, the white and red represent the unplasticized and plasticized particles, respectively. **b–d** velocity magnitude $|V| = \sqrt{v_x^2 + v_z^2}$ of slab's particles (colormap and arrow lengths) and direction of movement (arrow directions). **e–g** slope parallel strain rate. **h–j** Shear stress in the weak layer normalized by the gravity-induced vertical stress $\tau_0$, for each corresponding zoomed view above.

## Numerical experiments

Further insights were gained through numerical simulations. Here, we employ the 2D Material Point Method (MPM) model presented by Gaume et al.[37] (see Methods Section) to simulate crack propagation within an elastoplastic weak snow layer underlying a homogeneous, elastic snow slab (see also Supplementary Movie SM4) using field-measured parameters as model input in order to reproduce PST #3 (see Methods).

Our results (Fig. 5a) indicate that crack propagation occurs within the sub-Rayleigh regime, with speeds of approximately 50 m s$^{-1}$ for propagation distances typically below 5 m. In this regime, crack propagation is primarily driven by the collapse and subsequent bending of the slab, as illustrated in Fig. 5b. At a propagation distance of approximately 5.5 m, we observe the nucleation of a secondary (daughter) crack around 0.5 m ahead of the primary crack (Fig. 5b and Supplementary Fig. 8), suggesting the presence of the Burridge-Andrews mechanism[21,22]. This phenomenon is further supported by stress distributions within the weak layer: a single stress peak is observed in both the sub-Rayleigh (Fig. 5h) and supershear (Fig. 5j) regimes, whereas two distinct peaks emerge at the onset of daughter crack nucleation (Fig. 5i). To track the crack tip position in the weak layer (blue curve in Fig. 5a), we define it as the position of the farthest plasticized particle. This explains the apparent discontinuity in crack tip location at approximately 5.5 m of propagation, a feature that does not appear when tracking the crack tip through a threshold on the slab normal displacement (magenta curve). Beyond this transition, the crack enters a supershear propagation regime, reaching a speed of approximately 115 m s$^{-1}$ at a propagation distance of 8 m, consistent with experimental observations (Fig. 5a). However, our simulations suggest that steady-state crack propagation is not yet achieved at this point. Instead, a

steady-state regime is reached at around 15 m, where the formation of a Mach cone within the slab becomes evident (see Fig. 5d, 5g, as well as Supplementary Fig. 8 and Supplementary Movie SM4). Furthermore, we observe that the slab reaches a vertical displacement exceeding 50% of the total collapse height approximately 0.04 seconds after the crack tip has passed (Fig. 5a). Consequently, the crack speed evaluated using this vertical displacement threshold eventually reaches the supershear regime, but later than the crack itself, at a propagation distance of approximately 13 m. Finally, it is important to note that these findings are consistent with those obtained from entirely independent numerical simulations based on the Discrete Element Method (DEM), which models the weak layer as a highly porous, brittle thin layer using a simple bonded contact model (see Supplementary Note 4). Due to the simplicity of the contact laws in DEM simulations, they serve as true numerical experiments that, together with these continuum-based simulations, provide independent confirmation of our experimental findings and reinforce our conclusions.

## Discussion

The concept of supershear rupture was first theorized by Burridge[21] and Andrews[22]. It then took one decade to report on possible supershear earthquakes[38,39] and three decades to provide the first experimental validation[27]. Similarly, our motivation stems from recent numerical insights indicating a sub-Rayleigh to supershear transition in the release process of snow slab avalanches[20]. Although these findings were corroborated by indirect observations of minute changes in snow surface properties from avalanche videos, and more recently by DEM simulations[36], they nonetheless lacked experimental validation. The present study fills this gap by providing direct evidence of a supershear transition during a snow fracture test replicating avalanche release

conditions, shedding light on the deformation mechanisms associated with different propagation regimes. Our findings reconcile the shear versus collapse debate in avalanche science, showing that anticrack behavior dominates the early stages of failure (and/or in low-angle terrain), while shear-driven propagation takes over on steeper slopes after a few meters of crack growth.

Our experiments indicate that the transition to the supershear regime occurred at a distance of approximately 6 m from the initiation point. Hence, previous experiments were unable to capture this propagation mode and were limited to the sub-Rayleigh regime. Furthermore, while we focus on the experiment where a supershear fracture occurred, PST #1 and PST #2, conducted hours earlier, did not exhibit supershear speeds, providing important insights. In these two experiments, slab fractures disconnected the detached, sliding slab from the intact snowpack ahead of the crack tip. This limits the gravitational pull of the slipping slab and thus very likely prevented the transition to supershear crack propagation. Whether the absence of the slab fracture in PST #3 was caused by the natural strengthening[40] of the slab cannot be conclusively ascertained (see Supplementary Note 1). However, the slab had a very low density, and thus low strength, compared to typical slab layers in avalanches[41]. It can therefore be assumed that the transition to supershear crack propagation is not an exception in avalanche formation. In PST #3, a slab fracture also occurred close to the end of the column without arresting the crack in the weak layer. This aligns with the numerical findings of Trottet et al.[20], indicating that slab fractures do not stop the propagation once the supershear regime has set in. Crack arrest would then require heterogeneity of topography or mechanical properties[42]. The observation that supershear crack propagation is less likely stopped by slab fractures may have crucial practical implications. Similar to supershear earthquakes, which are generally associated with large magnitudes and propagation distances[43], supershear slab avalanches are expected to be associated with widespread crack propagation and extensive avalanche release areas, with significant consequences for risk management and mitigation. Conversely, in the sub-Rayleigh regime, slab fractures could either lead to "en-echelon" type fractures[44] or stop the crack propagation, thus leading to smaller release sizes.

It is important to highlight that the two key processes discussed here, anticrack and supershear mechanisms, were first introduced in earthquake science. Observations and reports of "whumpfs", firn and snowquakes[7,8], like deep earthquakes, are often associated with the volumetric collapse in porous deep layers and anticrack propagation[5,6]. In contrast, supershear slab avalanches are analogous to supershear strike-slip (mode II) earthquakes. Although these two regimes (anticrack and supershear) were discovered independently in seismology in two very different types of earthquakes, it is remarkable that they can occur sequentially during avalanche release. The primary difference between the fracture process in strike-slip earthquakes and snow slab avalanche release lies in the origin of the external loading: strike-slip earthquakes arise from stress build-up induced by tectonic plate motion, whereas supershear slab avalanches are driven by gravitational forces and stress accumulation near weak zones. Despite this difference in external loading, there are striking similarities in the underlying dynamics of both systems. For example, laboratory earthquake studies[45–47] have shown that the sub-Rayleigh regime is predominantly associated with fault-normal velocities, while the supershear regime is characterized by fault-parallel velocities. In a similar fashion, our results reveal that the anticrack sub-Rayleigh regime is marked by large slope-normal displacements and velocities, whereas the supershear regime is primarily associated with slope-parallel displacements and velocities.

Other mechanisms have been proposed to explain supershear fracture, such as transitions induced by barriers and asperities[48]. However, in our study, snow micro-penetrometer (SMP) measurements along the experimental snow field confirmed homogeneous conditions (see Supplementary Fig. 2). Furthermore, our numerical reproduction of the experiment using the Material Point Method (Fig. 5) clearly shows the nucleation of a daughter crack ahead of the main crack at a similar distance as in the field test (see also Supplementary Fig. 8). These observations strongly suggest that the supershear transition observed in our experiments is driven by the Burridge-Andrews mechanism. Finally, using the analytical expression for the supercritical crack length derived by Trottet et al.[20] under the assumption of the Burridge-Andrews mechanism, we estimated the supercritical crack length to range from 1.25 to 7.3 m based on a realistic weak layer shear strength (see Supplementary Note 4). This range aligns well with our experimental observations, further reinforcing our conclusion.

After the supershear transition, some studies have reported the presence of a trailing Rayleigh signature, a remnant of the main crack (the anticrack in our case). However, we did not observe such a trailing wave in either our experiments or numerical simulations. Our simulations suggest that this wave is quickly dissipated within the overall motion (see Supplementary Movie SM4). We observe that the slab flexural wave, which can be approximately tracked using a vertical displacement threshold (Fig. 5)[49], lags behind the supershear crack. Initially, in the sub-Rayleigh regime, slab flexure played a key role in driving crack propagation. However, after the supershear transition, crack propagation became governed primarily by slope-parallel tension. As a result, what was initially a flexural wave evolved into a secondary effect of shear failure and eventually reached intersonic speeds, though at a later stage, as evidenced by our numerical results.

A typical signature for supershear events at intersonic speeds is the observation of a Mach cone[27,47,50]. In our setup, the cone should be located inside the slab, starting at the crack tip. However, a Mach cone was not detected in our measurements. As shown numerically (Fig. 5, Supplementary Fig. 8, and Supplementary Movie SM4), steady-state supershear propagation would be required, together with high-resolution filming and speckling of the full slab depth, to precisely identify the Mach cone. Therefore, not detecting a Mach cone in our measurements could be due to a combination of factors: (i) the lower resolution of the full-view imaging compared to the close-up, (ii) the high-resolution close-up primarily focusing on the weak layer's vicinity rather than the entire slab depth to track the crack tip, and (iii) the propagation not yet reaching a steady state, making the cone formation and stabilisation unlikely. In the future, even longer experiments should be conducted to achieve a steady-state supershear regime and thus making the Mach cone visible in experiments. Additionally, wider tests are necessary to investigate the interplay between downslope/upslope and cross-slope crack propagation during supershear fractures. As increasing test size will naturally entail increased logistical workload and greater safety concerns, analyses based on fiber optics[51] installed in avalanche release zones could present an interesting and safe avenue for future research on supershear fracture mechanisms in snow slab avalanche release.

## Methods
### Snow fracture experiments
On 19 January 2021, we conducted a series of avalanche experiments in a natural snowpack on a small slope near Davos, Switzerland. To prevent natural avalanche release, or remote triggering of our study slope, when the weak layer was still forming at the snow surface, we compacted the weak layer at the perimeter of the planned experiments. During the subsequent snowfalls that buried the weak layer and formed the snow slab, we used snow saws about 1.2 m long to isolate the test site from the rest of the field to ensure that we would not fracture the weak layer while we worked around it. The experimental

setup is based on the Propagation Saw Test (PST), a standard fracture mechanical test for snow[35,52]. In the test, a column (width $b = 30$ cm) is isolated and an artificial cut is introduced into a pre-identified weak snow layer until a self-propagating crack starts when the critical cut length is reached. Our PSTs were around 10 m long, much longer than the 120 cm recommended standard length[32]. We performed three side-by-side PSTs within approximately 6 hours (PST #1, PST #2 and PST #3). The experiments were filmed with a Phantom VEO 710 high-speed (HS) camera. In PST #1, the HS camera was used to film the entire PST length (Fig. 1c, full-view), in PST #2 we filmed a close-up of the weak layer at a distance of 206–286 cm from the bottom edge of the PST, and in PST #3 we filmed a close-up at a distance of 861–919 cm. To also record the fracture event along the entire experiment, in PST #2 and PST #3 we filmed the entire PST with a standard digital camera (Sony RX100 V, 250 fps) (Fig. 1c, full-view).

## Snow properties

In close proximity to the PST experiment we characterized the snowpack by observing a traditional manual snow profile[31]. The weak layer consisted of buried surface hoar with a grain size of 10 - 15 mm below a 80 cm thick snow slab. The crack propagated at the bottom of this layer, at the transition to a 6 cm thick layer of faceted crystals resting above an ice crust (Supplementary Fig. 1). The slab above the weak layer mainly consisted of decomposing and fragmented precipitation particles and rounded grains, with a mean density of $102 \pm 5$ kg m$^{-3}$. Density was measured every 5 cm using a 100 cm$^3$ cylindrical density cutter (38 mm diameter). Spatial variability was documented with snow micro-penetrometer (SMP) measurements every 50 cm along the 10-meter-long PST. In general, the heterogeneity within the PST was relatively low (Supplementary Fig. 2), and the SMP measurements were in good agreement with the manual profile.

## Instrumentation

The recorded side wall of a PST was speckled with black ink (Indian Ink, Lefranc & Bourgeois) applied with a commercial garden pump sprayer. The HS camera recorded the experiments with a horizontal resolution of 1280 pixels and a frame rate of 5000 fps (full-view PST #1) or 10,000 fps (close-ups PST #2, PST #3). To avoid perspective distortion, we aligned the camera to be vertically and horizontally perpendicular to the wall and aimed the optical axis of the camera at the centre of the field of view, both horizontally and vertically. Correction for radial and tangential image distortion introduced by the camera lens was performed on the fly while correlating the images using the open source DICengine[53] software. Distortion coefficients $k_1$, $k_2$, $k_3$, $p_1$ and $p_2$ as well as the camera model were estimated using chessboard calibration images and is based on Bradski and Kaehler[54]. For a more thorough description of camera calibration we refer to Bergfeld et al.[33]. In the close-ups we used, in addition to the HS camera, a Sony camera (RX100 V) to capture PST #2 and PST #3 also in the full-view perspective. Movies were recorded at 250 fps and an image resolution of 1920 by 1080 pixels. In contrast to the raw images of the HS camera, the Sony camera recorded compressed MP4 files (h264-codec). Extracting individual images from the MP4 movie was done using FFMpeg[55]. Further camera settings can be found in Supplementary Table 1.

## Digital image correlation

DICengine was used for digital image correlation (DIC) analysis. It is an open source DIC tool to compute full-field displacements from sequences of digital images and it is provided by Sandia National Laboratories[53]. In the images, a region of interest (ROI) was selected encompassing the speckled PST wall. To derive displacement fields of the PST, the ROI was further subdivided into quadratic DIC subsets with a certain side length and step size (Supplementary Table 1). The position and deformation of each DIC subset was then tracked over time relative to an initial reference frame. In order to find the unique

DIC subsets in all subsequent frames, the DIC subsets can translate, rotate and deform through both normal and shear components. For an arbitrary DIC subset $i$, we thus obtained the initial horizontal and vertical position ($x_i$, $z_i$; for the definition of the coordinate system see Fig. 1c) as well as the time-dependent horizontal and vertical displacement $u_i(t)$ and $w_i(t)$ relative to this initial position.

## Post processing

Pixel-to-real-space conversion is done by manually picking the PST column length (or a ruler in the close-ups) as a reference to compute the pixel-to-meter conversion factor (Supplementary Table 1). To characterize the displacement behavior of the slab, we manually defined two time ranges: (i) a first time interval $\Delta t^{rest}$ when the slab-weak layer system was at rest, and (ii) the second time period $\Delta t^{crack}$ when crack propagation took place (Supplementary Fig. 2a and 3a).

The displacement $u_i(t)$ or $w_i(t)$ of a subset within $\Delta t^{rest}$ was characterized by two distinct noise sources. First, the signal exhibits high-frequency noise (Supplementary Figs. 2b, 3b, transparent lines), which is attributed to the electronic shot noise inherent in a camera sensor[56]. As the frame rate of the recorded videos is high compared to crack propagation time, we used a third-order Savitzky-Golay filter[57] to effectively reduce this high-frequency noise (Supplementary Figs. 2b, 3b, solid lines). Second, there were lower frequency sinusoidal oscillations in the HS-camera (not in the Sony camera). The sinusoidal noise comes from a built-in cooling fan and has a typical frequency of about 25 Hz. To remove this noise, the frequency and amplitude of the sinusoidal were fitted during $\Delta t^{rest}$ and suppressed for later times (Supplementary Fig. 2c, transparent and solid lines, respectively)).

In a second step, we applied tempo-spatial filtering to the displacement fields via Non-Local-Means (NLM[58]) video denoising (Supplementary Figs. 2d, e and 3c, d). In contrast to local filters, which smooth out sharp or discontinuous variations, the NLM method conserves fine detail while reducing additive noise[59–62]. The principle idea of NLM image denoising is to leverage the redundancy of information within a single image. Each image has small windows with similar patterns that repeat throughout the image. In NLM-denoising, a small window around a certain pixel is compared to all other windows within the neighborhood (or search domain $\Omega$). The similarity of the windows provides the weight of the neighborhood, while averaging the pixel in the center of the initial window. Very similar to repetitive patterns in one image, a video exhibits pattern repetition between frames. Hence, NLM-denoising can directly be extended from 2D (spatial) to 3D (tempo-spatial) denoising[63]. All experiments in this study were NLM–denoised with a window size $N = 3^3$ $pixel^3$, a search domain $\Omega = 21^3$ $pixel^3$, and a noise parameter $h$, which refers to the noise level of the signal of $h_X = 3 \times \sigma(X(\Delta t^{rest}))$. Here, $\sigma(X(\Delta t^{rest})$ represents the standard deviation of a signal $X$ in the time interval $\Delta t^{rest}$. Hence, the standard deviation of displacement $u(t)$ or $w(t)$ before cracking.

For the NLM algorithm, the measurement field cannot have any voids. However, due to the threshold-based subset initialization routine in DICengine, measurement points may be missing in areas of the ROI where speckling was poor (Supplementary Fig. 2d). As this poses a problem for the NLM algorithm, the voids were first filled using a contextual-recovery algorithm[64,65] based on the biharmonic equation.

From the filtered displacements of a subset, $u_i(t)$ and $w_i(t)$, velocity and acceleration were derived. This was done using the same Savietzky-Golay algorithm as for removing the high-frequency noise, but the first and second derivatives of the polynomial were considered. Strain fields were computed from differentiation of $u(t)$ and $w(t)$ with the second order accurate central differences[66].

In the last post-processing step, the ROI was divided into three subregions: (i) slab, (ii) weak layer and (iii) substrate. To define these subregions, we picked a time step at which strain concentrations in the weak layer were clearly visible (Supplementary Fig. 2f). The subregions

were then defined using manually selected thresholds for the strain. All DIC subsets above the upper boundary were assigned to the slab (Supplementary Fig. 2f, pink colored), all those below the lower boundary to the substrate (turquoise colored), and all in between to the weak layer.

The subset size in the full-view measurements was between 9.7 cm and 26 cm, in the close-ups it was 1.1 cm (close-up #1 performed in PST #2) and 1.7 cm (close-up #2 performed in PST #3). As weak layer thickness ranged from 4 to 8 mm, the subsets assigned to the weak layer also contain parts of slab and/or substratum. Measured strain is therefore always an effective quantity across the presumably smaller strain concentration within the weak layer. True measurements of strain in the weak layer are therefore not feasible with our video recordings.

## Crack speed evaluation

Measuring crack speed is essentially a question of identifying and following the crack tip with time. Compared to opening cracks when the material separates, for anticracks, this is more difficult as crack faces are not well defined[49]. In addition to snow crystal bond fracturing, new contacts also form during inter-penetration[25]. Hence, there is no clear state in which the crack flanks are clearly separated. Bergfeld et al.[33] compared three different methods to estimate crack speed. The first two were based on displacement and strain thresholds to estimate crack tip positions[67]. The third method tried to avoid the localization of the crack tip by considering the slab-weak layer system as translation invariant regarding the crack tip. In this method, the slab acceleration signal $\ddot{w}(t)$ was cross-correlated along the crack propagation path ($x$-direction in Fig. 1c) to estimate the time lag in which two locations along the column undergo a similar acceleration pattern. In summary, Bergfeld et al.[33] concluded that the correlation method is very sensitive to any dynamics during crack propagation. Crack speeds obtained with the correlation method were different compared to crack speeds derived via thresholds on the displacement or strain field. However, the methodological similarity between the latter two methods was also reflected in comparable crack speeds and trends along the propagation distance. Another work[49] applied a discrete micro-mechanical model to reproduce crack propagation dynamics and measure anticrack speed in numerical PSTs. Their analysis of weak layer stresses and bond breaking allowed them to monitor and track stress concentrations and the fracture process zone, also in transient crack propagation regimes. They applied four different methods to estimate crack speed. Besides threshold methods on slab displacement and acceleration, they also had access to stress and bond breakage in the weak layer. While their different methods yielded different crack tip locations, all methods provided very similar measures of crack speed. This finding further supports that displacement-based methods on experimental data to derive crack speed are appropriate. In this study, we largely followed the displacement-based idea from[33,67] to determine crack speed.

The deformation of the slab provides the energy to drive the crack in the weak layer, hence the deformation field of the slab is closely linked to the location and evolution of the crack tip. Different crack propagation modes result in different deformations of the slab. In mixed-mode anticrack the initial deformation is primarily driven by vertical displacement $w(t)$, while in the supershear regime, slab deformation is dominated by horizontal displacement $u(t)$ (see Fig. 4). Hence, for crack speed estimation of a mixed-mode anticrack the $w(t)$-field (Supplementary Fig. 6) and for cracks propagating in supershear mode the $u(t)$-field (Fig. 2) seem to be appropriate. This is particularly true when the fracture is not in a steady-state[49]. Since, the crack speed can therefore be based on $u(t)$ or $w(t)$, we now use $X$ as a placeholder:

The displacement $X_i$ of a subset $i$ located at the point $j = x$, $k = z$ and at a video frame $l = t$ is given by $X_{j,k,t}$. To determine crack speed, the displacement field $X_{j,k,t}$ was first averaged over the thickness of the slab.

$$\bar{X}_{j,l} = \frac{1}{N^z} \sum_k X_{j,k,t}$$

where $N^z$ is the number of vertically aligned subsets in the slab. From the displacement field $\bar{X}_{j,l}$, tempo-spatial plots can be drawn and give a good overview of the dynamics during crack propagation (Fig. 2d and Supplementary Fig. 5). At times ($y$-axis) and locations ($x$-axis) where the slab is not yet deformed we see a black region. For larger $x$-locations the black region extends to later times $t$, hence there is consecutive slab deformation due to cracking in positive $x$-direction.

For each location along the test column ($x$-axis), the time of cracking $t_c$, indicated by the accompanying deformation of the slab (transition from black to yellow), can be assessed. To identify the time of cracking $t_c(x)$ for each location $x$, we used a threshold of 25 times the standard deviation of displacement measured before cracking started ($t \in \Delta t^{rest}$). Crack speed was then measured as the slope of the linear regression in a least square manner of the data pairs ($t_c(x), x$) where $\{(t_c(x), x) | t \in \Delta t^{crack}, x_{\min} \leq x \leq x_{\max}\}$ and $x_{\min}, x_{\max}$ providing the range in which the crack speed is to be measured. In the example of Fig. 2d, the linear regression is the green line for the spatial sub-range 1 m ≤ $x$ ≤ 6 m and its slope represents the crack speed measure. To measure the evolution of crack speed along the PST column (see Fig. 2c) we used overlapping sub-ranges of twenty horizontally aligned subsets (≈ 1.2 m) with an overlap of 95%. The uncertainty in crack speed values was obtained by repeating the computation but changing thresholds to 50% and 150% of the nominal value, respectively.

## Elastic wave speed

Snow is a fragile, porous material whose mechanical behavior defies simple characterization. Due to the complex microstructure and the existence close to its melting point, measurements of the elastic modulus of snow presents numerous challenges (e.g., see Mellor[68]), resulting in published values that exhibit scatter over several orders of magnitudes. The observed scatter can be attributed to a number of factors, including differences in snow density as well as rate- and temperature-dependency. Moreover, snow microstructure as well as its anisotropy alters the elastic behavior[69,70].

In addition, crack propagation preceding an avalanche necessitates a layered snowpack, as the crack generally propagates in thin, weak layers. The energy required to create new crack surfaces (in the weak layer) originates from the slab layers above the weak layer, and the energy must be transferred towards the weak layer to further advance the crack. Consequently, there is an energy flux from the stress-strain field in the slab to the moving crack tip in the weak layer[71]. The relative magnitude of the strain components in the slab defines the fracture mode and gives rise to the associated wave, which transfers the energy towards the crack tip. Knowledge about elastic wave speeds is therefore crucial as these are bounding limits for the speed of the crack[72]. All wave velocity metrics directly depend on the material's elastic modulus, which poses two challenges: 1) which elastic modulus empirical parameterization should be used? and 2) what is the elastic wave speed in a layered slab? In this section, we address point 1) while point 2) is addressed in the Supplementary Note 3 by means of numerical modeling.

The difficulty in measuring the modulus of elasticity directly leads to uncertainty if an empirical density-to-modulus para- metrization is used to compute wave velocities. Therefore, shear wave speeds were computed based on five different published parametrizations[25,34,73–75] (Supplementary Fig. 7). For the measured mean slab density of $\rho = 102 \pm 5$ kg m$^{-3}$, this results in a Young's modulus ranging from 0.4 MPa to 1.5 MPa, and thus in shear wave speeds between 40 m s$^{-1}$ and 79 m s$^{-1}$. Despite the large spread in shear wave speed, measured supershear crack speeds were higher

than the highest values estimated from the slab density. In particular, the largest estimates from Sigrist[75] (dynamic cyclic loading at large strain rates) and Gerling et al.[34] (acoustic wave propagation measurements) were used as reference in the main text. The associated parameterizations are given by

$$E^{ger} = 25.4\,\text{MPa}\left(\frac{\rho}{\rho_{ice}}\right)^{4.6} \quad (1)$$

and

$$E^{sig} = 967\,\text{MPa}\left(\frac{\rho}{\rho_{ice}}\right)^{2.94}, \quad (2)$$

where $\rho_{ice}$ = 917 kg m$^{-3}$ is the density of ice[76].

### Numerical simulations

Numerical simulations where performed using the Material Point Method along with the cohesive Modified Cam-Clay (MCC) plastic model as introduced for snow slab avalanches by Gaume et al.[37] and used by Trottet et al.[20] to provide the first numerical evidence of supershear crack propagation in such avalanches. The snow is modeled with Hencky's finite strain elastic model to handle large deformations. The MCC plastic yield function and associated flow rule are added to the weak layer to account for fracture and allow crack onset and propagation. The elliptic yield function of the MCC allows for both volumetric and shear rupture, as well as strain softening and hardening. The slab is modeled using a hyperelastic model. The substratum is modeled as rigid. We refer to the previously cited articles[20,37] for more information about MPM and the cohesive MCC plastic model.

The simulated slab is homogeneous and has a density of $\rho$ = 102 kg m$^{-3}$ according to the measured mean density in the experiment. The Poisson's ratio is $\nu$ = 0.3. The Young's modulus was then derived using a dynamic experimental formula appropriate for the simulated dynamic regime. Sigrist's formula was preferred over Gerling's as it led to the highest elastic modulus prediction for our given density value (conservative estimate to ensure that supershear fracture occurred experimentally, see Fig. 2). All the simulation parameters are available in Supplementary Table 2).

## Data availability

All experimental data are available at the following public link https://doi.org/10.16904/envidat.679.

## Code availability

The MPM numerical model used in this study can be found in a previous publication at https://www.nature.com/articles/s41467-018-05181-w.

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

## Acknowledgements

We thank D. Kammer for general discussions on supershear fractures in geomaterials. Funding: This research was partly supported by the Swiss National Science Foundation (grant no. 200021_169424).

## Author contributions

Conceptualization: B.B., A.H., J.G., and J.S. Methodology: B.B., A.H., J.G. Experimental data acquisition and analysis: B.B., A.H.. Modeling and related analysis: A.P., G.B., and J.G. Visualization: B.B., G.B., J.G., A.P. Funding acquisition: A.H., J.S., and J.G. Supervision: J.G., A.H., and J.S. Writing: J.G. (Main Text), B.B. (Materials and Methods), B.B., A.P., G.B. (Supplementary Text) with contributions and revisions from J.S. and A.H.

## Competing interests

The authors declare no competing interests.
