## [Transparent Peer Review file · Nature Communications]

Signatures of the sub-Rayleigh to supershear fracture transition in snow avalanche experiments

Corresponding Author: Professor Johan Gaume

Version 0:

Reviewer comments:

Reviewer #1

(Remarks to the Author)

Let me start by saying that I find this work very interesting and exciting. This manuscript reports about the occurrence of supershear avalanches through experiments performed in a controlled outdoor environment. The experiments are conducted employing a well-established protocol, the propagation saw test (PST), where samples are obtained by sections cut in natural snowpack containing a weak layer buried under a cohesive thicker slab. By employing longer samples than in the standard PST, which allow for longer propagation distances, and by tracking the crack tip speed, the authors have been able to produce and observe supershear rupture propagation of the simulated snow avalanches, something that has never been observed before in laboratory experiments. I think this work provides an outstanding contribution to the field and I believe it would also be of interest to a broad readership.

I have the following questions and comments:

1. The authors claim that supershear transition occurs by the Burridge-Andrews transition mechanism but what evidence do they have for that? There are other mechanisms, including transition induced by asperities or barriers (e.g. Page, Dunham, Carlson, JGR 2005). How can they be sure that the Burridge-Andrews is the operating mechanism?
2. According to the Burridge-Andrews supershear transition mechanism, a secondary (supershear) crack nucleates ahead of the main (sub-Rayleigh) crack. After transition, there is still a trailing Rayleigh signature, a remnant of the main crack. Is this something that can be detected behind the supershear crack in the experimental measurements presented here?
3. Another way to demonstrate that the Burridge-Andrews is the operating mechanism would be to show that the supershear transition distance is consistent with the analytical estimate proposed by Trottet et al. Nature Physics 2022. If that is the case, does the supercritical crack length scale with the square root of the slab depth, consistent with Eq. 1 of Trottet et al.?
4. Conclusions are really based on a single test (PST#3). I am aware of the complexity involved in these tests. However, I think it would be important to check repeatability to add robustness to the findings. Also, I think it would be valuable to perform more experiments with slabs of different depths, which would allow to demonstrate, as mentioned above, the dependence of the supercritical crack length on the square root of the slab depth through the parameter Λ derived in Trottet et al. Nature Physics 2022.
5. PST#1 and PST#2 did not propagate to supershear speed because of slab fractures. It is interesting that a slab fracture also occurred in PST#3 but since the rupture had already transitioned to supershear speed, the slab fracture did not arrest it. How frequent are slab fractures that result in hindering supershear propagation? In other words, supershear avalanches are possible but how likely are they due to the effect of slab fractures? In the tests presented in this manuscript only 1 in 3. Again, I think that more tests would be helpful to answer this.
6. One of the main observations in this manuscript is that sub-Rayleigh propagation is characterized by a slope-normal collapse of the weak layer, whereas supershear propagation by predominantly slope-parallel motion. This observation is broadly consistent with the analytical solutions of Freund (1990) (and experimental measurements of Mello et al. Tectonophysics, 2011; Mello et al. JMPS, 2016; Rubino et al. JGR, 2020) for mode II cracks, where the interface-normal particle velocity predominates for sub-Rayleigh ruptures and interface-parallel predominates for supershear ruptures. To

what point do these similarities hold and what differences are there between these analytical solutions and experimental measurements and those presented here?

Some additional comments:

7. How are subset sizes chosen in the DIC analysis? Has a convergence study and error analysis been conducted to pick the right subset size? What is the error estimate on the presented measurements?

8. Material and Methods, page 16 (7th line from the bottom): is it not possible to select a smaller subset size so that the subset entirely falls within the weak layer and doesn't contain part of the slab and/or substratum.

9. What strategy has it been used to resolve discontinuous displacement fields across the interface of the weak layer with slab and substratum?

10. Material and Methods, page 16 (9th line from the top): the displacement field is averaged over the thickness of the slab. This averaging would be legitimate if the slab moved as a rigid body but is this the case? I would think that there are important displacement gradients through the thickness of the slab. In fact, there need to be such gradients to produce the heterogeneous strain rate fields of Extended Data Fig. 7.

11. What high-speed camera was used and what was the exposure time at 5000 and 10000 fps, respectively? Is it natural light enough even at the exposure time of 10000 fps or was any other light source employed?

12. Fig.3 and Extended Data Fig. 4: I would suggest adding and marking regions of close up # 1 and close up # 2 in panel (a), similar to supplementary movies 1 and 3.

13. Extended Data Fig.1: What are grain types a-e-d-g? Why not using the two-letter code indicated in "The international classification for seasonal snow on the ground" (ref. 30)? The grain type should be more clearly explained anyways.

Vito Rubino

Reviewer #2

(Remarks to the Author)

According to this manuscript by Bergfeld et al., confirmation of supershear cracks in snow was limited to avalanche videos (Hamre et al., 2014) and numerical simulations by the authors (Trottet et al., 2022). This next manuscript aims to re-confirm that their concept is valid, arguing that their earlier published experimental confirmation of supershear via full-scale measurements should be repeated in a controlled setting.

The manuscript describes that on 19 January 2021, a supershear crack was caught on camera in one fracture experiment. Since slab fracture did not arrest this shear crack, the authors also suggest it may lead to widespread propagation and believe this has "significant implications for risk mitigation strategies."

I don't know how experts in the Propagation Saw Test reject statistical flukes in a convincing way for statisticians. However, I can not encourage the publication of claims based on $n=1$. Currently, the evidence remains at a quite preliminary stage. From my own interaction with Nature Communications editors, I learned that they require a significantly more in-depth investigation for consideration as a full research article, and the journal does not have a brief communications format available as an alternative.

Moreover, even assuming that the experimental evidence is reproducible, I do not feel there are convincing analogies to earthquake phenomena (currently, even basic references to supershear strike-slip are confusing and not consistent with the authors' previous paper - Trottet et al., 2022 - as I explain below).

Also, the title is excellent clickbait but misleading: (i) no snowquake was studied in the paper, and (ii) I do not think avalanche professionals call the Propagation Saw Test a laboratory avalanche. According to my reading of the manuscript, the title is more like: "One propagation saw test reveals supershear crack."

Finally, some sections of the manuscript read like a remake of Trottet et al., 2022, but while some sentences feel like Deja Vu, others contradict the author's previous words, and some alignment of arguments and the storyline seem necessary.

DETAILED REMARKS (in the absence of line numbers, I just copy-paste text segments):

p. 2, suspected in firm and snowquakes [7,8,9]

I am unsure how thermal contraction cracks studied in [8] relate to the anticrack.

p. 2, limited by the Rayleigh wave speed

A reference would be helpful.

p. 3-4, laboratory (here and elsewhere)

To me, this is an “outdoor experimental site”, because a laboratory is “a room or building with equipment for doing scientific tests” (<https://dictionary.cambridge.org/>).

p. 4, supershear snowquake (here and elsewhere)

I do not see any rationale to make “intersonic fracture” a synonym of “supershear snowquake” except just making it sound cool. Mars quake, ice quake, Moon quake, etc., refer to a seismic event or vibration produced by something, like a crack. No vibrations were recorded to declare there was any kind of quake, and I am not even sure we know if it is detectable for different phases of the rupture.

p. 5, Fig. 2

I understand the rationale later appearing in Methods on p. 18 about $w(t)$, $u(t)$. Nevertheless, I find [a-Normal, close-up #1] next to [b-Tangential, close-up #2] confusing. Comparison of the same components seems more straightforward. It makes the figure speak for itself without needing a reader to pause and figure out what is going on and whether the normal component drifts to subpanel (c) and how it falls into the context of shear in (d). This figure is introduced with a discussion of tangential displacement analysis, and only half a page later, the reader is informed that normal tangential displacements in close-up #1 yield similar speeds without these shown on the plots. Perhaps, caption could clarify that $c = 41$ m/s in (a) was also obtained from $u(t)$.

c^{sig} , c^{ger} could be explained in the caption.

p. 6, normal displacements differed.

What was the speed based on the latter?

p. 6, aligns with the assumption

It was not introduced earlier and might need a reference.

p. 8, It is noteworthy

It could be helpful to explain why it is so because it may not be evident to the reader.

p. 9, “providing direct evidence of a supershear snowquake during snow avalanche release”

I do not think any evidence related to the snowquake was provided; i.e., no measurements showed that the supershear crack is a seismic event.

The present study does not provide evidence “during snow avalanche release,” only “during one Propagation Saw Test.”

p. 10

It is unclear how the unlikely crack arrest by slab fracture “parallels findings in earthquake science,” and should be supported with references.

I could not follow the discussion about the crack arrest. On the one hand, Fig. 4 shows that normal collapse is uncoupled from tangential and arrives with a delay to the 8.5 m camera position. If so, vertical slab fracture is unlikely to take place before arrival of normal displacement, which lags behind the supershear, and thus cannot arrest the crack. Moreover, I am confused about how Fig. 4 aligns with the numerical findings of Trottet et al. (2022), suggesting that anticrack propagation is not relevant at distances larger than 3-5 m (Trottet et al., 2022).

“supershear earthquakes are generally associated with large magnitudes” needs a reference and specification of which magnitude. Moreover, in Trottet et al. (2022), the same authors claim the opposite: “supershear propagation has rarely been reported in large strike-slip earthquakes.”

Anyway, I do not understand the “Just as” comparison of earthquakes to “supershear snowquakes” and wider release areas. Do supershear earthquakes have a large size because they rupture fast? Also, in Trottet et al. (2022), the larger avalanche size was associated with higher slab density, making fracture arrest unlikely, and I am unsure how the two effects coexist.

“Our findings reinforce the analogy between avalanches and earthquakes.” I could not understand which analogy was reinforced? The anticrack concept inspired by deep-earthquakes, which can be completely neglected by using simple shear models (Trottet et al. 2022)?

I could not follow the logic here: “firn and snowquakes, like deep earthquakes, are often associated with the volumetric

collapse ... and anticrack propagation. In this context, supershear snowquakes are analogous to supershear strike-slip (mode II) earthquakes." First, are supershear strike-slip earthquakes often associated with volumetric collapse and anticracks? I do not think so. Second, are supershear strike-slip earthquakes represent deep earthquakes? Again, I do not think so, because strike-slip earthquakes are usually shallow.

In Discussion, I missed implications for the debate motivating the paper.

p. 30, crust arrow
Im not sure where does it point vertically.

p. 32, typo
of of the crack tip.

p. 36
I think h-a-b layerings labels should be explained,
Similarly, $\rho_{\{m,-,+}}$ could be associated to ED table 2

p. 39, _
theclose-up

Reviewer #3

(Remarks to the Author)

The authors report results of an experimental study of slab avalanche release. The experiments are denoted as laboratory experiments which is an interesting terminology given that they were conducted on a natural slope in natural snowpack (I can understand why the authors prefer not to speak of field experiments). They adhere to the protocol of the propagation saw test, a standard testing method designed to probe anticrack-type failure of collapsible weak layers in snow. The performed high-speed imaging and DIC analysis yield high-quality data for the displacement fields which indicate, albeit only in one of three samples, a transition from subshear anticrack to supershear crack propagation. In two other samples the same transition was not observed because of slab rupture after an otherwise similar anticrack propagation stage.

We find this study, which seems to partly confirm an earlier theoretical prediction by some of the authors, to be interesting and potentially important in view of our understanding of the various factors that decide the size of slab avalanches. The fact that not only the fracture propagation speed but also the slip pattern changes in the late stage of crack propagation adds to the evidence for a qualitative transition in propagation mechanism. The authors provide a good but not exhausting documentation of their data. In particular, we would have liked to see more data and clarification on the following points:

- 1) The authors mention that crack speeds determined from u and w coincide for the 'anticrack' stage but differ for the 'supershear' stage. Can you give the respective c_u and c_w vs x curves or equivalent information?
- 2) Displacement fields are shown for the 'successful' experiment but not for its two 'unsuccessful' counterparts. It would be important to know whether in these two experiments the 'anticrack' type displacement pattern persists throughout the propagation. Can you show these fields either in form of supplementary videos or graphics?
- 3) In extended figure 7, it might be good to see a 'zoom' at about 10 m, i.e. near the location of the experimental window, to clearer assess the expected existence or not of a mach cone at that location. Also, despite strenuous efforts we could not find a definition for the abbreviation 'FOV'. This has not helped our understanding.
- 4) The authors propose that a critical crack length is required for a transition into the supershear regime. A related definition of critical crack length for supershear was provided in a prior study using numerical simulations (Trottet, B., Simenhois, R., Bobillier, G. et al., Nat. Phys. 18, 1094–1098 (2022)), which is also cited in this paper. It would be interesting to see if this critical crack length can be linked to the observations.
- 5) The Burridge–Andrews mechanism is alluded to multiple times in the paper. While the analogy with strike-slip earthquakes is clear, how exactly the Burridge-Andrews mechanism (initiation of a daughter crack in front of the main crack) takes place in this case (especially PST #3) is unclear in the paper.

Overall, we think that this paper may make an important contribution of general interest, if the mentioned points can be fully addressed.

Reviewer #4

(Remarks to the Author)

Version 1:

Reviewer comments:

Reviewer #1

(Remarks to the Author)

I think the authors have done a great job at revising the manuscript. The remaining weak point is the repeatability of the supershear event. While reporting a single supershear event falls short of providing evidence for repeatability, I also think that due to the challenges involved in performing such "field experiments", it is not realistic to produce experiments at the same rate as in a fully controlled laboratory environment.

The authors explain this very clearly in their response letter, discussing the requirement for a "very unstable snowpack – specifically, a persistent weak layer prone to crack propagation – combined with a safe slope for testing" and saying that "such conditions are seldom met simultaneously". However, I think that these complexities are not emphasized as much in the manuscript. I believe it may be valuable to add some of this description somewhere in the method's section, in order to illustrate the specific challenges of these field experiments that set them apart from tests conducted on lab specimens, especially considering the broader readership of this article.

The numerical simulations with the material point method and the discrete element method well complement the experimental observations and add robustness to the study. Overall, I think this manuscript can provide a very valuable contribution and I recommend it for publication.

I also have some minor suggestions:

- The term "Supershear fracture" is often used in the manuscript. It is general in nature and it probably fine. At the same time, I may also suggest perhaps a slightly more appropriate term: "supershear rupture" as this is typically used in the literature to refer to (frictional) ruptures along pre-defined interfaces, as it is the case for those propagating along the weak layer. This is opposed to fractures breaking bonds in a material or along a cohesive interface.
- Fig. 3a: add label with close up#1 and #2 to connect the text more easily with the figure, without having to refer to Fig. 1.
- Fig. 5: Magnitude of speed -> Velocity magnitude

Reviewer #2

(Remarks to the Author)

This is my second review of the study, and I find it in much better shape than the initial submission. The conclusions are still based on n=1 tests, but considering the demanding nature of such tests and in combination with numerical simulations, are convincing enough for publication.

I have little to add, except some technical remarks:

Abstract:

"Here, we report direct observations of a supershear fracture in snow avalanche release following the spontaneous transition from sub-Rayleigh to intersonic crack propagation." - This is an overstatement as no direct observations were made in snow avalanche release, only in a propagation saw test (or snow fracture test).

Fig. 5b-d:

"IVI" could be defined in the caption.

Reviewer #3

(Remarks to the Author)

We are happy that the authors have comprehensively addressed the issues raised in our previous review. The added material is significant and has clarified our questions. Also the changes made in the title have helped to clarify the scope and content of the manuscript. Finally we note that the somewhat misleading term 'laboratory experiment' has been changed. We think that the paper provides an important contribution to the general literature, notably in view of the existence of similar phenomena in earthquakes. Also the obtained data are of very high quality and will be of wide interest to researchers involved in avalanche modelling. We recommend the paper for publication.

Reviewer #4

(Remarks to the Author)

Response to reviewer's comments

Manuscript NCOMMS-24-62974-T

Signatures of the sub-Rayleigh to supershear fracture transition in snow avalanche experiments
by Bergfeld, Gaume, Pellet, Bobillier, Schweizer, van Herwijnen

In the following, we provide (in blue) detailed point-by-point answers to the comments raised by the reviewers (in black, *italic*). In addition, modifications made to the manuscript are highlighted in a separate file with tracked-changes.

Response to Referee #1 – Prof. Dr. Vito Rubino

Let me start by saying that I find this work very interesting and exciting. This manuscript reports about the occurrence of supershear avalanches through experiments performed in a controlled outdoor environment. The experiments are conducted employing a well-established protocol, the propagation saw test (PST), where samples are obtained by sections cut in natural snowpack containing a weak layer buried under a cohesive thicker slab. By employing longer samples than in the standard PST, which allow for longer propagation distances, and by tracking the crack tip speed, the authors have been able to produce and observe supershear rupture propagation of the simulated snow avalanches, something that has never been observed before in laboratory experiments. I think this work provides an outstanding contribution to the field and I believe it would also be of interest to a broad readership.

We thank Referee #1 for his positive comments on our manuscript and for very constructive suggestions that helped us to improve the readability and quality of our paper (see detailed replies below).

I have the following questions and comments:

1. *The authors claim that supershear transition occurs by the Burrige-Andrews transition mechanism but what evidence do they have for that? There are other mechanisms, including transition induced by asperities or barriers (e.g. Page, Dunham, Carlson, JGR 2005). How can they be sure that the Burrige-Andrews is the operating mechanism?*

Thank you for this interesting reference. It is particularly relevant for cases where crack propagation is influenced by macro asperities, such as in forested terrain or in the presence of large obstacles. However, in our study, no such asperities were present. We verified through SMP (Snow Micro Penetrometer) measurements along the beam that we had homogeneous conditions (See Extended Data Figure 1). Given this, we assumed that the supershear transition follows the Burrige-Andrews mechanism. While we do not have high-speed imagery at the precise location of the transition, our numerical simulations using DEM and MPM closely reproduce the experimental conditions and provide strong evidence for the Burrige-Andrews mechanism (see Figures 1, 7 and 5 in this rebuttal letter which were also added to our manuscript and supplement). Specifically, we observe the nucleation of a daughter shear crack ahead of the main crack at a location consistent with the observed supershear transition, supporting this interpretation (Fig. 1). We have clarified in the manuscript that our interpretation is based on this assumption, supported by numerical experiments (and directly by the field tests) that are now also presented in the main text.

2. *According to the Burrige-Andrews supershear transition mechanism, a secondary (supershear) crack nucleates ahead of the main (sub-Rayleigh) crack. After transition, there is still a trailing Rayleigh signature, a remnant of the main crack. Is this something that can be detected behind the supershear crack in the experimental measurements presented here?*

After the supershear transition, Rubino et al. (2020) have indeed reported the presence of a trailing Rayleigh signature, reminiscent of the main crack (the anticrack in our case). However, we did not observe such a trailing wave in either our experiments or numerical simulations. Our simulations suggest that this signature is visible for a few milliseconds after the transition but that it is quickly dissipated within the overall motion (see

Figure 1: Numerical experiment of weak layer crack propagation beneath a homogeneous snow slab using a 2D Material Point Method. (a) Time according to crack tip position, determined by following either the plasticized particles in the weak layer (blue) or the slab's slope perpendicular collapse (magenta). The grey dotted line represents a virtual crack tip moving at the speed of the shear waves. (b to g) Zoomed views of the slab (colormapped) and the weak layer (white and red) around the crack tip. In the weak layer, the white and red represent the unplasticized and plasticized particles respectively. b-d : magnitude of speed (colormap and arrow lengths) and direction of movement (arrow directions). e-g : slope parallel strain rate. (h to j) Shear stress in the weak layer normalized by the gravity induced vertical stress τ_0 , for all three zoom views above.

Figure 2: Evidence of the Burridge Andrews mechanism in MPM simulation of the PST 3 experiment. Plasticized particles in the weak layer are colored in red and the slab is colored by the longitudinal rate of deformation (left) and shear rate of deformation (right). The supershear transition occurs at approximately 5.5 m, consistent with the transition reported in PST 3. A Mach cone begins to form beyond this point, becoming distinctly visible once a steady state is achieved. See Figure 1.

Figure 2 in this letter and Supplementary Movie 4). Yet, we observe that the slab flexural wave, which can be approximately tracked using a vertical displacement threshold (Fig. 1) [1], lags behind the supershear crack. Initially, in the sub-Rayleigh regime, slab flexure played a key role in driving crack propagation. However, after the supershear transition, fracture propagation became governed by slope-parallel tension. As a result, what was initially a flexural wave evolved into a secondary effect of shear failure and eventually reached intersonic speeds, though at a later stage. We have added a discussion about this interesting aspect in the Discussion section.

3. Another way to demonstrate that the Burridge-Andrews is the operating mechanism would be to show that the supershear transition distance is consistent with the analytical estimate proposed by Trottet et al. *Nature Physics* 2022. If that is the case, does the supercritical crack length scale with the square root of the slab depth, consistent with Eq. 1 of Trottet et al.?

Thank you for this suggestion. Let us recall the formula for the supercritical crack length from Trottet et al.:

$$a_{sc} = \Lambda \frac{\tau_p - \tau_g}{\tau_g - \tau_r} \quad (1)$$

where $\Lambda = \sqrt{E' D D_{wl} / G_{wl}}$ (E' is the plane strain elastic modulus, D and D_{wl} are the slab and weak layer thicknesses, respectively, and G_{wl} is the shear modulus of the weak layer). Additionally, $\tau_g = \rho g D \sin \psi$ represents the shear stress induced by the slab load, $\tau_r = \mu_d \rho g D \cos \psi$ is the residual frictional resistance, and $\tau_p = c + \mu \sigma_n$ denotes the shear strength of the weak layer. Note that μ_d represents the dynamic (or crack face) friction, while μ is the internal friction coefficient. The cohesion of the weak layer can therefore be calculated using the following equation:

$$c = \frac{a_c}{\Lambda} (\tau_g - \tau_r) + \tau_g - \mu \sigma_n \quad (2)$$

This calculation is particularly useful as it allows for a direct comparison between values obtained through back-calculation and shear frame measurement data from the literature. Assuming $\mu_d = 0.6$ [2], $\mu = 0.36$ [3], $F_{wl} = D_{wl} / G_{wl} = 10^{-7}$ m/Pa [4], and using the elastic modulus evaluation from [5], we obtain a cohesion value of $c = 1.4$ kPa (assuming $a_{sc} = 5.5$ m). This falls within the range of values measured by [6] and [7]. Conversely, performing the inverse calculation by assuming a realistic range of cohesion values between 0.5 and 2 kPa yields a range of supercritical crack lengths between 1.25 and 7.3 m. While this analysis does not provide a single definitive prediction due to uncertainties in weak layer strength, it strengthens the conclusions of our paper. It was included in the Discussion section and a Supplementary Note. Furthermore, we conducted an additional analysis using two independent numerical methods (MPM and DEM, see Figures 1 and 5) to verify whether the field experiments could be reproduced in terms of crack speed regimes, deformation patterns, and supercritical crack length. The results of this analysis, presented in this letter and also in the revised manuscript, also help address other questions raised by this reviewer and other reviewers.

4. Conclusions are really based on a single test (PST#3). I am aware of the complexity involved in these tests. However, I think it would be important to check repeatability to add robustness to the findings. Also, I think it would be valuable to perform more experiments with slabs of different depths, which would allow to demonstrate, as mentioned above, the dependence of the supercritical crack length on the square root of the slab depth through the parameter Λ derived in Trottet et al. *Nature Physics* 2022.

We appreciate the reviewer's comment highlighting the need for repeatability and additional experiments. While we agree on the importance of robustness in scientific findings, we emphasize the exceptional challenges involved in conducting supershear experiments in snow. Performing long Propagation Saw Tests (PSTs) under the necessary conditions is extremely rare and logistically complex. These experiments require a very unstable snowpack—specifically, a persistent weak layer prone to crack propagation—combined with a safe slope for testing. Such conditions are seldom met simultaneously, but in January 2021, we identified a rare opportunity at a low-elevation site in Davos where these conditions were finally satisfied.

One significant challenge is that the weak layer must be carefully preserved during setup to avoid pre-triggering or natural/remote avalanche triggering. To prepare the slope, we removed the weak layer at the perimeter of

the planned experiments while it was still forming at the snow surface. After a subsequent large snowfall, we ensured the weak layer remained intact and stable until the experiments were conducted. This combination of factors is exceedingly rare, as highly unstable snowpacks are typically found at higher-altitude alpine sites, where the logistical and safety challenges of performing such experiments are significantly greater. Over several years, we have made repeated attempts to conduct long PST experiments, but these efforts were largely unsuccessful due to these constraints.

In January 2021, however, we successfully performed several long PST experiments at the Davos site. In one of these tests, we observed supershear fracture, conclusively supported by high-resolution, full-field Digital Image Correlation (DIC) analysis and close-up views, which independently confirmed the same behavior. These multiple data points, including continuous speed estimates and localized measurements, provide robust, high-quality evidence for the occurrence of supershear fracture, leaving no ambiguity about the results.

Regarding the suggestion to investigate the dependence of the supercritical crack length on slab depth, we agree this is an interesting direction for future research. However, performing additional field experiments to validate this dependence would likely require numerous years due to the rarity of favorable conditions. To address this, we have complemented our fieldwork with numerical experiments, such as simulations using Material Point Method (MPM) and Discrete Element Method (DEM), which offer a more practical and feasible approach to explore this phenomenon.

Finally, we note that many foundational studies on supershear fractures, such as those examining supershear earthquakes, are based on single-event observations. While repeatability is ideal, our findings are supported by comprehensive, high-resolution data as well as additional numerical experiments and contribute valuable insights into the dynamics of supershear fractures.

5. PST#1 and PST#2 did not propagate to supershear speed because of slab fractures. It is interesting that a slab fracture also occurred in PST#3 but since the rupture had already transitioned to supershear speed, the slab fracture did not arrest it. How frequent are slab fractures that result in hindering supershear propagation? In other words, supershear avalanches are possible but how likely are they due to the effect of slab fractures? In the tests presented in this manuscript only 1 in 3. Again, I think that more tests would be helpful to answer this.

This is an important point. We have shown in Trotter et al. and in Fig. 3 below that tensile failure of soft slab can prevent the supershear transition (panels a and c). In this case, the volumetric collapse of the weak layer drives anticrack propagation because the slab cannot accumulate the necessary amount of strain energy required for supershear fracture. This leads to the so-called ‘en-echelon’ type of fracture. The crack propagation speed in this case is varying within a sub-Rayleigh range (panel e). The propagation speed decreases locally after the onset of slab tensile failure. Eventually, crack propagation may be stopped after an ultimate slab fracture. In contrast, for harder slabs, slab tensile failure may occur but does not influence crack propagation dynamics and supershear fracture (panels b and d). In that case, the crack propagation speed is almost identical to that obtain for an elastic slab. Therefore, we assume that supershear fracture requires relatively hard slabs. In the context of our experiments, in PST 1 and PST 2, we argue that the slab was still too soft to allow the transition towards supershear fracture. In PST 3, the absence of the slab fracture was very likely caused by the natural strengthening of snow which is known to occur over very short timescales [8]. This strengthening effect is particularly efficient and fast in the case of new snow with low initial density.

Figure 3: Effects of slab fracture on the supershear transition (for a slope angle of 30 degrees). From MPM simulations [9] in which we account for slab tensile failure for a relatively soft ($\sigma_t = 6$ kPa, left) and hard ($\sigma_t = 11$ kPa, right) slab. a, b: PST colored by normalized vertical displacement (slab) and weak layer fracture (in red) at two different times. c, d: spatio-temporal evolution of h/D_{wl} and crack tip location. Black solid lines: crack tip position; empty circles: critical crack length a_c ; filled circles: super critical crack length a_{sc} . Red lines: slab fracture. e, f: normalized propagation speed.

6. One of the main observations in this manuscript is that sub-Rayleigh propagation is characterized by a slope-normal collapse of the weak layer, whereas supershear propagation by predominantly slope-parallel motion. This observation is broadly consistent with the analytical solutions of Freund (1990) (and experimental measurements of Mello et al. Tectonophysics, 2011; Mello et al. JMPS, 2016; Rubino et al. JGR, 2020) for mode II cracks, where the interface-normal particle velocity predominates for sub-Rayleigh ruptures and interface-parallel predominates for supershear ruptures. To what point do these similarities hold and what differences are there between these analytical solutions and experimental measurements and those presented here?

We thank the reviewer for these insightful references. While our findings are qualitatively consistent with the results of these studies, specifically, that slope-normal velocities dominate in the sub-Rayleigh regime and slope-parallel velocities in the supershear regime (Figure 4), quantitatively applying the theoretical framework proposed by Mello et al. to our setup is more complex. There are significant differences between our experimental configuration and the laboratory earthquake experiments. First, our model involves an additional factor: gravity, which influences both slope-normal and slope-parallel velocities. These velocities are directly affected by gravitational acceleration, as well as by weak layer compaction energy, collapse height, and friction. This combination of factors prevents a direct application of the theoretical framework developed by Mello et al. Furthermore, another key difference lies in the experimental setup itself. In laboratory earthquake experiments, homogeneous samples are typically pre-stressed under a predefined loading angle, allowing precise control over crack front speed and enabling more targeted and repeatable measurements. In contrast, we are observing crack propagation in a naturally layered snowpack, where supershear propagation arises from a continuously evolving stress state within the system, rather than from externally imposed conditions. We have added sentences to the Discussion section to elaborate on these differences and include the referenced studies.

Figure 4: Velocity direction of supershear (red) and subrayleigh (blue) closeup.

Some additional comments:

7. How are subset sizes chosen in the DIC analysis? Has a convergence study and error analysis been conducted to pick the right subset size? What is the error estimate on the presented measurements?

Automated subset initialization was performed using feature matching. This process required specifying the subset size, step size, and a threshold for the Sum of Square of Subset Intensity Gradients (SSSIG). SSSIG, which quantifies speckle pattern quality, guided the selection of subset size by starting with a small value and progressively increasing it until exceeding the predefined threshold (see Extended Data Table 1). This procedure was applied separately for each camera and field of view. To estimate errors (uncertainties), we predicted the standard deviation of the displacement solution based on the estimated image noise and SSSIG [10]. For the high-speed camera capturing close-up views, typical values were around 0.01 mm. Since this uncertainty is specific to the DIC analysis, we also computed an alternative displacement uncertainty after postprocessing (see Materials and Methods). This was determined by calculating the standard deviation of displacements at times identified as preceding cracking (Extended Data Figure 2a, light gray background). Both error estimates are included in Extended Data Table 1.

8. Material and Methods, page 16 (7th line from the bottom): is it not possible to select a smaller subset size so that the subset entirely falls within the weak layer and doesn't contain part of the slab and/or substratum.

Unfortunately not. The black-white contrast of the speckle pattern is insufficient for this purpose. Applying and optimization of speckle pattern presents several challenges. Snow is a porous material, and its crystal size is comparable to that of the weak layer thickness. Individual crystals reflect light, complicating the process. Additionally, the snow is just below freezing point, while the black paint is above it when it is applied. Different speckle patterns must be applied for varying fields of view and camera setups. Even seemingly minor issues, such as frozen spray nozzles, can cause significant problems in the field. Furthermore, the black ink drastically alters energy input from solar radiation. After application, the blackened snow surface rapidly heats up, melts, and destroys the speckle pattern within minutes. Since test samples are naturally formed and snowpack stability changes quickly (within hours to days), test repetition is impossible. Whilst we agree that it would be beneficial if the size of the subset was smaller than the thickness of the weak layer, we have experienced great difficulty in getting the subsets smaller in field experiments.

9. What strategy has it been used to resolve discontinuous displacement fields across the interface of the weak layer with slab and substratum?

Essentially, we removed regions around the discontinuity after analysis by attributing them to the weak

layer, thereby keeping the slab and substratum free from larger errors and poor correlation quality. To further prevent the smearing of the discontinuity (beyond the inherent smoothing effect of the subset size) we applied Non-Local Means (NLM) filtering, which provides excellent denoising while preserving edges and features, such as crack discontinuities.

We assume your comment was more focused on specialized strategies for handling crack-induced discontinuities in DIC (e.g., Regularized Finite Element-Based DIC or subset splitting). While we did not apply such techniques, we recognize specific challenges for anticracks: (i) Subset Interference (In anticracks, displacement fields converge, leading to overlapping subsets instead of a clean separation); (ii) Loss of Speckle Uniqueness (In an anticrack, speckles may move toward each other or overlap, breaking tracking consistency); (iii) Strain Field Ambiguities (For anticracks, strain fields often appear as high compression zones, making it harder to define a clean "split" for subset splitting).

10. Material and Methods, page 16 (9th line from the top): the displacement field is averaged over the thickness of the slab. This averaging would be legitimate if the slab moved as a rigid body but is this the case? I would think that there are important displacement gradients through the thickness of the slab. In fact, there need to be such gradients to produce the heterogeneous strain rate fields of Extended Data Fig. 7.

Extended Data Fig. 7 (c, e, f) presents strain rate fields derived from MPM simulations. The red-outlined region marks the field of view of the close-up recordings from the experiments. Within this region, minimal heterogeneity is observed in the z-direction, which is also reflected in the experimental results (Extended Data Fig. 2d, e; Extended Data Fig. 3c, d; Extended Data Fig. 4d).

In Extended Data Fig. 4e, the bending-induced tangential displacement of the anticrack exhibits a displacement gradient. However, from the perspective of the static camera, the displacement field shifts uniformly with crack propagation. Therefore, we consider averaging over the thickness to be a valid approach for assessing crack speed, as long as the deformation field remains largely self-similar with increasing crack propagation distance, hence strong dynamics are not present. That said, we acknowledge that this assumption does not hold at the supercritical crack length, where the transition to supershear propagation occurs. However, at even larger crack propagation distances, the deformation fields appear to regain self-similarity with propagation distance. In this context, we would like to draw attention to Supplementary Movies SM2 and SM3, which show the time-resolved displacement fields and their homogeneity.

11. What high-speed camera was used and what was the exposure time at 5000 and 10000 fps, respectively? Is it natural light enough even at the exposure time of 10000 fps or was any other light source employed?

A Phantom VEO 710 high-speed camera was used (see Materials and Methods). Exposure times were $199.6 \mu\text{s}$ and $99.0 \mu\text{s}$, respectively. The frame rate was limited by the available daylight. Consequently, the first experiment was filmed at 5000 fps, as direct sunlight was not available at that time of day. We added the missing information in the text and Extended Data Table 1.

12. Fig.3 and Extended Data Fig. 4: I would suggest adding and marking regions of close up # 1 and close up # 2 in panel (a), similar to supplementary movies 1 and 3.

Thank you for this suggestion, we have changed the figures accordingly.

13. Extended Data Fig.1: What are grain types a-e-d-g? Why not using the two-letter code indicated in "The international classification for seasonal snow on the ground" (ref. 30)? The grain type should be more clearly explained anyways.

Thanks for pointing this out. We have switched to the two-letter code and provided further explanation in the caption

Response to Referee #2

We thank Referee #2 for their insightful comments and suggestions, which helped identify areas where our discussion lacked clarity and could be misinterpreted. These remarks have been very helpful in improving both the readability and overall quality of our manuscript. Below, we provide detailed responses addressing each point raised.

According to this manuscript by Bergfeld et al., confirmation of supershear cracks in snow was limited to avalanche videos (Hamre et al., 2014) and numerical simulations by the authors (Trottet et al., 2022). This next manuscript aims to re-confirm that their concept is valid, arguing that their earlier published experimental confirmation of supershear via full-scale measurements should be repeated in a controlled setting.

The manuscript describes that on 19 January 2021, a supershear crack was caught on camera in one fracture experiment. Since slab fracture did not arrest this shear crack, the authors also suggest it may lead to widespread propagation and believe this has “significant implications for risk mitigation strategies.”

I don't know how experts in the Propagation Saw Test reject statistical flukes in a convincing way for statisticians. However, I can not encourage the publication of claims based on $n=1$. Currently, the evidence remains at a quite preliminary stage. From my own interaction with Nature Communications editors, I learned that they require a significantly more in-depth investigation for consideration as a full research article, and the journal does not have a brief communications format available as an alternative.

We respectfully disagree with the assessment that our evidence is preliminary. Our conclusions are supported by high-resolution, full-field Digital Image Correlation (DIC) analysis and close-up views, both of which independently confirm the same behavior. These multiple data sources—including continuous speed estimates and localized measurements—provide robust, high-quality evidence for the occurrence of supershear fracture, leaving no ambiguity in our results.

We fully recognize the importance of robust scientific findings and are confident in the robustness of our results. Regarding additional potential measurements, we emphasize the exceptional challenges of conducting supershear experiments in snow. Long Propagation Saw Tests (PSTs) under the required conditions are exceedingly rare due to the necessity of a highly unstable snowpack with a persistent weak layer prone to crack propagation, combined with a safe testing environment. Such conditions are seldom met simultaneously. In January 2021, we identified a unique opportunity at a low-elevation site in Davos, where these conditions were finally satisfied.

A major difficulty is preserving the weak layer during setup to avoid premature triggering. To mitigate this, we carefully removed the weak layer at the perimeter of the planned experiments while it was still forming at the snow surface. After a subsequent large snowfall, we ensured it was intact and stable until testing. These conditions are exceptionally rare, as highly unstable snowpacks are typically found at higher altitudes, where logistical and safety constraints make controlled experiments nearly impossible. Despite repeated attempts over several years, we were unable to conduct successful long PSTs until this rare opportunity arose.

During this campaign, we successfully performed several long PST experiments, and in one of them, we observed supershear fracture, conclusively validated by high-resolution, full-field DIC analysis and close-up views. The consistency across multiple measurements reinforces the reliability of our findings.

It is also worth noting that many foundational studies on supershear fractures—including those examining supershear earthquakes—are based on single-event observations. While repeatability is ideal, our findings are underpinned by comprehensive, high-resolution data that offer new insights into supershear fracture dynamics.

To further substantiate our results, we conducted two independent numerical simulations using the Discrete Element Method (DEM, discrete approach) and the Material Point Method (MPM, continuum approach) to replicate the experimental setup. Both approaches independently led to the exact same conclusions, further strengthening the robustness of our experimental findings. The results of the MPM simulation are presented

in Figure 1, which has been included in the main paper. The results of numerical experiments using DEM are shown below but were not included in the main paper, as they produce essentially the same output as the MPM simulations while requiring a significantly expanded Methods section. It was thus included as Supplementary Note instead. Both simulations were conducted using the same geometric setup for the weak layer-slab system, with homogeneous slab properties (density and elastic modulus) matching the experimental data (average values) and a weak layer shear strength within the measured range reported by Jamieson and Johnson [6] (adjustable parameter). Both simulations were able not only to reproduce the observed experimental data very well in terms of sub-Rayleigh speed, supershear speed, supercritical crack length as well as deformation patterns. They also offered detailed insights into the transition mechanism, following, according to both DEM and MPM simulations the Burridge Andrews mechanism (see Figures 1c, 1i, 5c.2). These numerical experiments further substantiate our experimental findings and reinforce the robustness of our conclusions. In particular, it fully rules out any potential "statistical flukes" as suggested by the reviewer, as our observations are supported not only by high-resolution data from multiple sources but also by results from two independent numerical models.

In addition, we recognize (see also below) that the term "Laboratory avalanche" in the title was not the best choice, as it might have misleadingly suggest that these experiments are easily reproducible. In reality, our study represents a rare and exceptional achievement, capturing unprecedented insights into avalanche release and fracture mechanics. The title was therefore modified accordingly.

Moreover, even assuming that the experimental evidence is reproducible, I do not feel there are convincing analogies to earthquake phenomena (currently, even basic references to supershear strike-slip are confusing and not consistent with the authors' previous paper - Trottet et al., 2022 - as I explain below).

As explained below, we acknowledge that a particular sentence in Trottet et al. may be open to misinterpretation. We will clarify our intended meaning and make the necessary revisions in the text of the present paper to prevent any misunderstandings. In addition, following the comments of Reviewer 1, we explored the earthquake literature in greater depth and identified further similarities with our experimental results, most notably, the observation that sub-Rayleigh crack propagation is dominated by slope- or fault-normal displacement, whereas supershear propagation is characterized by dominant slope- or fault-parallel displacement (Freund (1990), Mello et al. Tectonophysics, 2011; Mello et al. JMPS, 2016; Rubino et al. JGR, 2020).

Also, the title is excellent clickbait but misleading: (i) no snowquake was studied in the paper, and (ii) I do not think avalanche professionals call the Propagation Saw Test a laboratory avalanche. According to my reading of the manuscript, the title is more like: "One propagation saw test reveals supershear crack."

We initially used the term "snowquake," which is often associated with anticrack-related "whumpfs," to emphasize the dynamic nature of the fracture process observed in our experiments, which shares similarities with seismic events in terms of energy release and rapid crack propagation. However, we agree that this could be misleading, and we have removed the term "snowquake" from both the title and abstract. We have retained its use in the introduction, where we describe the "whumpf" sounds as well as anticracks and snowquakes reported in the snow literature. Regarding "laboratory avalanches", in seismology, it is common to refer to basic shear tests as "laboratory earthquakes" due to their ability to replicate certain features of natural seismic events. Similarly, a Propagation Saw Test (PST) reproduces several key ingredients of a snow slab avalanche, making it reasonable to refer to it as a "laboratory avalanche." Nevertheless, we understand that this term may be misleading, especially in the context of the previous remark, and we are happy to modify the title to better reflect the scope and focus of our study.

Finally, some sections of the manuscript read like a remake of Trottet et al., 2022, but while some sentences feel like Deja Vu, others contradict the author's previous words, and some alignment of arguments and the storyline seem necessary.

If any sentences appear to contradict the findings of Trottet et al., this must have been an unintended oversight on our part and we will make the necessary changes to improve the storyline here (see below).

Figure 5: DEM simulation of the PST 3 experiment. The first row shows time versus crack tip location in DEM and MPM simulations. The second row presents slab displacement fields, where color indicates displacement norm (maximum value 6 mm) and arrows represent displacement direction, shown in the sub-Rayleigh regime, at the onset of the supershear transition, and during steady-state supershear propagation. The third row displays broken bonds, while the fourth and fifth rows show normal and shear stress distributions in the weak layer, respectively. Results indicate that in the sub-Rayleigh regime, the crack tip remains sharply defined and aligns closely with the peak in normal stress. At the transition, new bond breakage occurs significantly ahead of the main anticrack, which is still driven by normal stress, whereas these newly broken bonds correspond to the peak in shear stress. In the supershear regime, the crack tip sharpens again and aligns with the shear stress peak.

DETAILED REMARKS (in the absence of line numbers, I just copy-paste text segments):

p. 2, suspected in firm and snowquakes [7,8,9] I am unsure how thermal contraction cracks studied in [8] relate to the anticrack.

Thank you for pointing this out. The reference [8] was indeed not relevant to the anticrack discussion, and we have removed it from the manuscript.

p. 2, limited by the Rayleigh wave speed A reference would be helpful.

We added reference to Siron et al. [11].

p. 3-4, laboratory (here and elsewhere) To me, this is an “outdoor experimental site”, because a laboratory is “a room or building with equipment for doing scientific tests” (<https://dictionary.cambridge.org/>).

For geologists and environment engineers, the field is often referred to as the laboratory (in Latin, combination between *laborare*: to work and *-orium*: place associated with a specific activity). We used this term to distinguish between “classical” avalanche dynamics experiments performed at a very large scale. Despite this, we modified the terminology accordingly throughout the whole paper and now refer to snow avalanche experiments.

p. 4, supershear snowquake (here and elsewhere) I do not see any rationale to make “intersonic fracture” a synonym of “supershear snowquake” except just making it sound cool. Mars quake, ice quake, Moon quake, etc., refer to a seismic event or vibration produced by something, like a crack. No vibrations were recorded to declare there was any kind of quake, and I am not even sure we know if it is detectable for different phases of the rupture.

Although it is certain that such a fracture phenomenon generates a seismic signal, we agree that the seismic analysis was not the topic of the study. The term snowquake was removed from the title, abstract and from most of the paper (except in the introduction / state of the art section where we believe it is important to mention).

p. 5, Fig. 2 I understand the rationale later appearing in Methods on p. 18 about $w(t)$, $u(t)$. Nevertheless, I find (a-Normal, close-up #1) next to (b-Tangential, close-up #2) confusing. Comparison of the same components seems more straightforward. It makes the figure speak for itself without needing a reader to pause and figure out what is going on and whether the normal component drifts to subpanel (c) and how it falls into the context of shear in (d). This figure is introduced with a discussion of tangential displacement analysis, and only half a page later, the reader is informed that normal tangential displacements in close-up #1 yield similar speeds without these shown on the plots. Perhaps, caption could clarify that $c = 41$ m/s in (a) was also obtained from $u(t)$.

We followed the reviewer’s suggestion and added more details to the figure caption to highlight how the speeds indicated in each subplot were evaluated.

c^{sig} , c^{ger} could be explained in the caption.

This was added as suggested.

p. 6, normal displacements differed. What was the speed based on the latter?

The w -based speed was about 80 m/s. This can also now be seen more clearly in the new plot Extended Data Fig. 5 (see Figure 6 below) in which the speed evaluation in the full-view is shown both based on w and u displacements. A similar analysis was also added to the main text in a new figure incorporating the results

from numerical experiments reproducing the field tests.

Figure 6: Slope-normal displacement with time for the three PST experiments (a,b,c). The slope-parallel displacement is shown at the bottom (e,f,g). Slab fractures are shown with red arrows pointing at x -location they appeared.

p. 6, aligns with the assumption

It was not introduced earlier and might need a reference.

We have added a reference to Trottet et al. [9]

p. 8, It is noteworthy It could be helpful to explain why it is so because it may not be evident to the reader.

We added to the text that this observation could be expected since fracture propagation in this regime is driven by weak layer collapse and therefore slope-normal displacement.

p. 9, “providing direct evidence of a supershear snowquake during snow avalanche release” I do not think any evidence related to the snowquake was provided; i.e., no measurements showed that the supershear crack is a seismic event.

This was reworded as discussed above.

The present study does not provide evidence “during snow avalanche release,” only “during one Propagation Saw Test.”

We have reformulated that we obtained evidence of a supershear transition during a snow fracture test replicating avalanche release conditions.

p. 10

It is unclear how the unlikely crack arrest by slab fracture “parallels findings in earthquake science,” and should be supported with references.

The misunderstanding likely arose from the following sentence on our part, which we agree is unclear: *'The observation that supershear crack propagation is less likely stopped by slab fractures, which parallels findings in earthquake science, may have crucial practical implications. Just as supershear earthquakes are generally associated with large magnitudes, supershear snowquakes resulting in slab avalanches are expected to come along with widespread crack propagation and extensive avalanche release areas, with significant consequences for risk management and mitigation.'* What we meant is that, in avalanches with supershear fractures, crack propagation is less likely to be halted by slab fractures. This suggests that supershear avalanches may involve widespread crack propagation, leading to large avalanche dimensions. The association between supershear avalanches and large dimensions parallels findings in earthquake science, as supershear earthquakes have only been observed in cases of large-magnitude events. We have clarified this important aspect in the revised text.

I could not follow the discussion about the crack arrest. On the one hand, Fig. 4 shows that normal collapse is uncoupled from tangential and arrives with a delay to the 8.5 m camera position. If so, vertical slab fracture is unlikely to take place before arrival of normal displacement, which lags behind the supershear, and thus cannot arrest the crack. Moreover, I am confused about how Fig. 4 aligns with the numerical findings of Trottet et al. 2022, suggesting that anticrack propagation is not relevant at distances larger than 3-5 m (Trottet et al., 2022).

This is not entirely accurate and we clarify below. Slab fractures are not only due to normal collapse. Slab fractures can be caused by slab bending (induced by normal collapse) indeed or by slab tension (induced by slope parallel deformation) or a combination of the two. What we say here is that in the supershear shear regime, slab tension is building up and may lead to slab fracture. However, it was shown by Trottet that these slab fractures (purely induced by slab tension and not by collapse) could not stop the supershear fracture. We would like to point the reviewer towards the supplement in Trottet et al. and Figure 3 in this document showing simulations including slab fractures. We recall them below for clarity. In the supershear regime, the collapse wave still exists but is lagging behind even for large propagation distances. However it represents a secondary process and does not drive the crack anymore. In addition, please note that while the slab tensile stress induced by bending is limited by the extent of collapse, the tensile stress induced by slope-parallel tension continues to increase as the supershear crack propagates. Consequently, slab fracture will eventually occur; however, our numerical results, supported by the experiment, demonstrate that this slab fracture does not halt supershear crack propagation in the weak layer. Therefore, in this regime, we expect topography to be the primary factor determining where the crack stops and, consequently, the size of the avalanche release.

“supershear earthquakes are generally associated with large magnitudes” needs a reference and specification of which magnitude. Moreover, in Trottet et al. (2022), the same authors claim the opposite: “supershear propagation has rarely been reported in large strike-slip earthquakes.”

We have added a reference to Bao et al. [12]. Regarding Trottet, it seems the sentence was misinterpreted, and we apologize if the main message was not clear. What we intended to convey is that supershear fractures have only been reported in very large and therefore rare cases of strike-slip earthquakes. Thus, both claims align and point in the same direction. We will ensure this is clarified in the revised text.

Anyway, I do not understand the “Just as” comparison of earthquakes to “supershear snowquakes” and wider release areas. Do supershear earthquakes have a large size because they rupture fast? Also, in Trottet et al. (2022), the larger avalanche size was associated with higher slab density, making fracture arrest unlikely, and I am unsure how the two effects coexist.

The sentence has been modified and clarified. Regarding avalanches, yes, many observations [13] show that the thicker and denser the slab is, the larger is the avalanche release zone. Here, we assume that denser slabs have a larger tensile strength and are therefore less likely to break before the supershear transition is met. For instance after 5 meters of propagation, on a 35 degrees slope, the tension in a slab of density 250 kg m⁻³ would be around 2 kPa whereas the tensile strength would be around 3 - 3.5 kPa according to Jamieson [14]. Therefore, enough tension can build up without slab fracture for the onset of supershear fracture. We have expanded

the discussion on this important aspect.

“Our findings reinforce the analogy between avalanches and earthquakes.” I could not understand which analogy was reinforced? The anticrack concept inspired by deep-earthquakes, which can be completely neglected by using simple shear models (Trottet et al. 2022)?

Although we report that on a steep slope the anticrack mechanism is no longer the main driver of the propagation, it is still an important concepts that explains ”whumps” as well as remote avalanche triggering. The concept of anticrack was first identified in earthquake science and later recognized as relevant to snow avalanches. Similarly, supershear fracture has been extensively studied in earthquake science to explain the exceptionally high fracture speeds observed in some large-magnitude strike-slip earthquakes. In this paper, we present direct observations of supershear fractures in snow avalanche experiments, which we believe strengthen the analogy between earthquakes and snow slab avalanches. Furthermore, while anticrack behavior is typically associated with deep earthquakes and supershear fractures with strike-slip earthquakes, our findings show that both mechanisms can coexist and transition within a single slab avalanche. Although the coexistence of these regimes within a single event has not been reported in earthquake science, it is remarkable that these two processes (anticrack and supershear fracture), discovered independently in seismology, can occur sequentially during avalanche release. This was clarified.

I could not follow the logic here: “firn and snowquakes, like deep earthquakes, are often associated with the volumetric collapse . . . and anticrack propagation. In this context, supershear snowquakes are analogous to supershear strike-slip (mode II) earthquakes.” First, are supershear strike-slip earthquakes often associated with volumetric collapse and anticracks? I do not think so. Second, are supershear strike-slip earthquakes represent deep earthquakes? Again, I do not think so, because strike-slip earthquakes are usually shallow.

We trust that the revisions to the text have clarified this point, as the reviewer’s interpretation does not accurately reflect our intended meaning.

In Discussion, I missed implications for the debate motivating the paper.

This aspect was brought back to the Discussion.

p. 30, crust arrow Im not sure where does it point vertically.

It points towards the very thick crust layer which appears as a high-density line just below the arrow.

p. 32, typo of of the crack tip.

Thank you. This was corrected.

p. 36 I think h-a-b layerings labels should be explained, Similarly, $\rho_{m,-,+}$ could be associated to ED table 2

This is now better explained in the figure caption.

p. 39, theclose-up

Thanks for catching this error.

Response to Referees #3 and #4

The authors report results of an experimental study of slab avalanche release. The experiments are denoted

as laboratory experiments which is an interesting terminology given that they were conducted on a natural slope in natural snowpack (I can understand why the authors prefer not to speak of field experiments). They adhere to the protocol of the propagation saw test, a standard testing method designed to probe anticrack-type failure of collapsible weak layers in snow. The performed high-speed imaging and DIC analysis yield high-quality data for the displacement fields which indicate, albeit only in one of three samples, a transition from subshear anticrack to supershear crack propagation. In two other samples the same transition was not observed because of slab rupture after an otherwise similar anticrack propagation stage.

We find this study, which seems to partly confirm an earlier theoretical prediction by some of the authors, to be interesting and potentially important in view of our understanding of the various factors that decide the size of slab avalanches. The fact that not only the fracture propagation speed but also the slip pattern changes in the late stage of crack propagation adds to the evidence for a qualitative transition in propagation mechanism. The authors provide a good but not exhausting documentation of their data. In particular, we would have liked to see more data and clarification on the following points:

We thank Referees #3 and #4 for their positive comments on our manuscript and for very constructive suggestions that helped us to improve the readability and quality of our paper (see detailed replies below).

1) The authors mention that crack speeds determined from u and w coincide for the 'anticrack' stage but differ for the 'supershear' stage. Can you give the respective c_u and c_w vs x curves or equivalent information?

We have now included several additional plots in the suggested direction. Figure 6 in this document has been added as an Extended Data Figure, illustrating crack speed evaluation for the three PST cases based on both u and w displacements. Additionally, we compare the crack tip position to the location where the slab reaches 50% of the collapse height—a metric often used to characterize the flexural wave in previous studies (e.g., Gaume et al. [15]). This comparison is presented in both the experiments and the numerical simulations in Fig. 5 of the paper, as well as in Figure 1 of this document.

2) Displacement fields are shown for the 'successful' experiment but not for its two 'unsuccessful' counterparts. It would be important to know whether in these two experiments the 'anticrack' type displacement pattern persists throughout the propagation. Can you show these fields either in form of supplementary videos or graphics?

Thank you for this suggestion. To respond to this comment and other remarks by another reviewer, we have added Extended Data Figure 5 which compares the full-view analysis for the three PSTs including the cases in which supershear fracture did not occur. When it comes to the displacement fields in the other tests, we have now included 3 new supplementary videos showing the displacement vectors for the three PSTs based on the full-view analysis, as suggested.

3) In extended figure 7, it might be good to see a 'zoom' at about 10 m, i.e. near the location of the experimental window, to clearer assess the expected existence or not of a mach cone at that location. Also, despite strenuous efforts we could not find a definition for the abbreviation 'FOV'. This has not helped our understanding.

To address this comment, we have added a new plot to the paper as Extended Data Figure 8 which serves the purpose of illustrating both the Burridge-Andrews mechanism and the apparition of the Mach cone. At the location of close-up 2, the Mach cone is visible but not perfectly clear yet because the steady state has not been reached yet. We have also reworded FOV field of view.

4) The authors propose that a critical crack length is required for a transition into the supershear regime. A related definition of critical crack length for supershear was provided in a prior study using numerical simulations (Trottet, B., Simenhois, R., Bobillier, G. et al., *Nat. Phys.* 18, 1094–1098 (2022)), which is also cited in this paper. It would be interesting to see if this critical crack length can be linked to the observations.

Figure 7: Evidence of the Burridge Andrews mechanism in MPM simulation of the experiment. Plasticized particles in the weak layer are colored in red and the slab is colored by the longitudinal rate of deformation (left) and shear rate of deformation (right). The supershear transition occurs at approximately 5.5 m, consistent with the transition reported in . A Mach cone begins to form beyond this point, becoming distinctly visible once a steady state is achieved.

In a new Supplementary Note (#4), we estimate the expected supercritical crack length based on weak layer shear strength measurements by Jamieson and Johnson [6]. Using a realistic shear strength range of 0.5 to 2 kPa, we obtain supercritical crack lengths between 1.25 and 7.3 m, aligning with experimental results. While this analysis does not yield a single definitive prediction due to uncertainties in snow property measurements, it reinforces the conclusions of our study. This discussion has been incorporated into the main text. Furthermore, we conducted an additional analysis using two independent numerical methods (continuum: MPM and discrete: DEM) to verify whether the field experiments could be reproduced in terms of crack speed regimes, deformation patterns, and supercritical crack length. The results of this analysis, presented in Figure 1, also help address other questions related to the Burrige-Andrews mechanism and Mach cone existence.

5) *The Burrige–Andrews mechanism is alluded to multiple times in the paper. While the analogy with strike-slip earthquakes is clear, how exactly the Burrige-Andrews mechanism (initiation of a daughter crack in front of the main crack) takes place in this case (especially PST #3) is unclear in the paper.*

We hope that the inclusion of numerical simulations, additional illustrations, and text modifications will help clarify this important point. Notably, we have added a dedicated section in the Results discussing the evaluation and interpretation of the Burrige-Andrews mechanism in PST 3, based on a numerical reproduction of this test.

Overall, we think that this paper may make an important contribution of general interest, if the mentioned points can be fully addressed.

Thank you very much.

References

- [1] Grégoire Bobillier, Bastian Bergfeld, Jürg Dual, Johan Gaume, Alec van Herwijnen, and Jürg Schweizer. Micro-mechanical insights into the dynamics of crack propagation in snow fracture experiments. *Scientific Reports*, 11(11711), 2021.
- [2] A van Herwijnen and J. Heierli. Measurements of crack-face friction in collapsed weak snow layers. *Geophys. Res. Lett.*, 36, 2009. L23502.
- [3] I. Reiweger, J. Gaume, and J. Schweizer. A new mixed-mode failure criterion for weak snowpack layers. *Geophys. Res. Lett.*, 42(5):1427–1432, 2015.
- [4] Bettina Richter, Jürg Schweizer, Mathias Rotach, and Alec van Herwijnen. Validating modeled critical crack length for crack propagation in the snow cover model SNOWPACK. *The Cryosphere Discussions*, (June):1–21, 2019.
- [5] C Sigrist. *Measurements of fracture mechanical properties of snow and application to dry snow slab avalanche release*. PhD thesis, ETH Zürich, 2006.
- [6] J.B Jamieson and C.D Johnston. Evaluation of the shear frame test for weak snowpack layers. *Ann. Glaciol.*, 32:59–69, 2001.
- [7] B. Jamieson and J. Schweizer. Texture and strength changes of buried surface-hoar layers with implications for dry snow-slab avalanche release. *J. Glaciol.*, 46:151–160, 2000.
- [8] Denes Szabo and Martin Schneebeli. Subsecond sintering of ice. *Applied Physics Letters*, 90(15):151916, 2007.

- [9] Bertil Trottet, Ron Simenhois, Gregoire Bobillier, Bastian Bergfeld, Alec van Herwijnen, Chenfanfu Jiang, and Johan Gaume. Transition from sub-Rayleigh anticrack to supershear crack propagation in snow avalanches. *Nature Physics*, 18(9):1094–1098, 7 2022.
- [10] Bing Pan, Huimin Xie, Zhaoyang Wang, Kemaq Qian, and Zhiyong Wang. Study on subset size selection in digital image correlation for speckle patterns. *Opt. Express*, 16(10):7037–7048, May 2008.
- [11] Marin Siron, Bertil Trottet, and Johan Gaume. A theoretical framework for dynamic anticrack and supershear propagation in snow slab avalanches. *Journal of the Mechanics and Physics of Solids*, 181:105428, 2023.
- [12] Han Bao, Liuwei Xu, Lingsen Meng, Jean-Paul Ampuero, Lei Gao, and Haijiang Zhang. Global frequency of oceanic and continental supershear earthquakes. *Nature Geoscience*, 15(11):942–949, 2022.
- [13] Johan Gaume, G. Chambon, N. Eckert, M. Naaim, and J. Schweizer. Influence of weak layer heterogeneity and slab properties on slab tensile failure propensity and avalanche release area. *Cryosphere*, 9(2):795–804, 2015.
- [14] B Jamieson and C Johnston. In-situ tensile tests of snowpack layers. *J. Glaciol.*, 36(122):102–106, 1990.
- [15] J. Gaume, A. van Herwijnen, G. Chambon, K. Birkeland, and J. Schweizer. Modeling of crack propagation in weak snowpack layers using the discrete element method. *The Cryosphere*, 9:1915–1932, 2015.

Response to reviewer's comments

Manuscript NCOMMS-24-62974-T

Signatures of the sub-Rayleigh to supershear fracture transition in snow avalanche experiments
by Bergfeld, Gaume, Pellet, Bobillier, Schweizer, van Herwijnen

In the following, we provide (in blue) detailed point-by-point answers to the comments raised by the reviewers (in black, *italic*). In addition, modifications made to the manuscript are highlighted in a separate file with tracked-changes.

Response to Referee #1 – Prof. Dr. Vito Rubino

I think the authors have done a great job at revising the manuscript. The remaining weak point is the repeatability of the supershear event. While reporting a single supershear event falls short of providing evidence for repeatability, I also think that due to the challenges involved in performing such “field experiments”, it is not realistic to produce experiments at the same rate as in a fully controlled laboratory environment.

The authors explain this very clearly in their response letter, discussing the requirement for a “very unstable snowpack – specifically, a persistent weak layer prone to crack propagation – combined with a safe slope for testing” and saying that “such conditions are seldom met simultaneously”. However, I think that these complexities are not emphasized as much in the manuscript. I believe it may be valuable to add some of this description somewhere in the method’s section, in order to illustrate the specific challenges of these field experiments that set them apart from tests conducted on lab specimens, especially considering the broader readership of this article.

The numerical simulations with the material point method and the discrete element method well complement the experimental observations and add robustness to the study. Overall, I think this manuscript can provide a very valuable contribution and I recommend it for publication.

We thank Referee #1 for his positive comments on our manuscript and his recommendation for publication. To address the comment about the experimental setup difficulties, we have added the following sentence at the beginning of the results section, as suggested: *Supershear experiments in snow are extremely challenging, as they require a highly unstable snowpack and a safe, accessible slope, conditions that rarely coincide. Even under favorable conditions, long propagation tests rarely succeed, especially for distances of 10 meters or more [1]. The successful experiments presented here followed years of unsuccessful attempts and extensive site preparation to preserve the weak layer and ensure favorable test conditions.*

I also have some minor suggestions:

- The term “Supershear fracture” is often used in the manuscript. It is general in nature and it probably fine. At the same time, I may also suggest perhaps a slightly more appropriate term: “supershear rupture” as this is typically used in the literature to refer to (frictional) ruptures along pre-defined interfaces, as it is the case for those propagating along the weak layer. This is opposed to fractures breaking bonds in a material or along a cohesive interface.

We thank the referee for this comment. While we understand that “supershear rupture” is commonly used in the context of frictional ruptures along pre-existing interfaces, our study involves cohesive-frictional interfaces, where bond breaking at the grain scale (modeled here through plasticity theory) plays a central role. As such, the process we describe is not purely frictional but involves the propagation of a fracture front through a weak layer. For this reason, we believe that the term “supershear fracture” remains appropriate in our context and better reflects the underlying physical processes.

- Fig. 3a: add label with close up1 and 2 to connect the text more easily with the figure, without having to refer to Fig. 1.

This was modified as suggested.

- Fig. 5: Magnitude of speed – > Velocity magnitude

This was corrected as suggested.

Response to Referee #2

This is my second review of the study, and I find it in much better shape than the initial submission. The conclusions are still based on $n = 1$ tests, but considering the demanding nature of such tests and in combination with numerical simulations, are convincing enough for publication.

We thank Referee 2 for their positive feedback and are pleased that our revisions were convincing enough to warrant their recommendation for publication.

I have little to add, except some technical remarks: Abstract: "Here, we report direct observations of a supershear fracture in snow avalanche release following the spontaneous transition from sub-Rayleigh to intersonic crack propagation." - This is an overstatement as no direct observations were made in snow avalanche release, only in a propagation saw test (or snow fracture test).

We modified this sentence and now refer to snow fracture experiments.

Fig. 5b-d: "IVI" could be defined in the caption.

This is now defined in the caption.

Response to Referee #3

We are happy that the authors have comprehensively addressed the issues raised in our previous review. The added material is significant and has clarified our questions. Also the changes made in the title have helped to clarify the scope and content of the manuscript. Finally we note that the somewhat misleading term 'laboratory experiment' has been changed. We think that the paper provides an important contribution to the general literature, notably in view of the existence of similar phenomena in earthquakes. Also the obtained data are of very high quality and will be of wide interest to researchers involved in avalanche modelling. We recommend the paper for publication.

We thank Referee #3 for their positive comments on our manuscript and their recommendation for publication.

Response to Referee #4

We thank Referee #4 for their positive comments on our manuscript and their recommendation for publication.

References

- [1] EH Bair, R Simenhois, A van Herwijnen, and K Birkeland. The influence of edge effects on crack propagation in snow stability tests. *The Cryosphere*, 8:1407–1418, 2014.